# Vangl2 promotes the formation of long cytonemes to enable distant Wnt/β-catenin signaling

Lucy Brunt[1], Gediminas Greicius[2], Sally Rogers[1], Benjamin D. Evans ⬤ [1,3], David M. Virshup ⬤ [2], Kyle C. A. Wedgwood ⬤ [1] & Steffen Scholpp ⬤ [1✉]

Wnt signaling regulates cell proliferation and cell differentiation as well as migration and polarity during development. However, it is still unclear how the Wnt ligand distribution is precisely controlled to fulfil these functions. Here, we show that the planar cell polarity protein Vangl2 regulates the distribution of Wnt by cytonemes. In zebrafish epiblast cells, mouse intestinal telocytes and human gastric cancer cells, Vangl2 activation generates extremely long cytonemes, which branch and deliver Wnt protein to multiple cells. The Vangl2-activated cytonemes increase Wnt/β-catenin signaling in the surrounding cells. Concordantly, Vangl2 inhibition causes fewer and shorter cytonemes to be formed and reduces paracrine Wnt/β-catenin signaling. A mathematical model simulating these Vangl2 functions on cytonemes in zebrafish gastrulation predicts a shift of the signaling gradient, altered tissue patterning, and a loss of tissue domain sharpness. We confirmed these predictions during anteroposterior patterning in the zebrafish neural plate. In summary, we demonstrate that Vangl2 is fundamental to paracrine Wnt/β-catenin signaling by controlling cytoneme behaviour.

[1] Living Systems Institute, School of Biosciences, College of Life and Environmental Sciences, University of Exeter, Exeter, UK. [2] Program in Cancer and Stem Cell Biology, Duke-NUS Medical School, Singapore, Singapore. [3] School of Psychological Science, Faculty of Life Sciences, University of Bristol, Bristol, UK. ✉email: s.scholpp@exeter.ac.uk

Long-distance cell-cell communication is essential for development and function of multicellular organisms. During embryogenesis, shape-forming signals, called morphogens, are produced at a localized source and act both locally and at a distance to control morphogenesis. In a concentration-dependent manner, morphogens orchestrate the cellular fates in their signaling range by controlling the gene expression of key transcription factors[1–3]. A tightly regulated distribution of morphogens is a prerequisite to allowing their precise spatial and temporal function during embryogenesis in tissue patterning and organ development.

Morphogens of the Wnt signaling family are a class of secreted ligands, that can transduce their signals through several distinct pathways to regulate a diverse array of developmental processes[4,5]. The best-characterized Wnt pathway is the Wnt/β-catenin dependent signaling pathway[6]. Wnt ligands together with Frizzled receptors and the co-receptors Lrp5/6 stabilize the key downstream target β-catenin. The co-transcription factor β-catenin, together with TCF/Lef transcription factors, mediates many cellular processes such as cell differentiation and proliferation and determines tissue patterning along the anteroposterior (AP) body axis. The planar cell polarity (PCP) signaling pathway is a β-catenin-independent pathway in the Wnt signaling network[7,8]. The Wnt/PCP pathway regulates cytoskeleton remodeling by activation of c-Jun N-terminal kinases (JNK) and members of the Rho-family GTPases, such as Cdc42 and RhoA to direct cellular morphogenesis, tissue polarity, and cell migration[9]. Within a cell, β-catenin-dependent and PCP-dependent Wnt signaling are well known to act in a mutually repressive manner and inhibiting one will typically upregulate the other.

In vertebrates, the transduction of Wnt signaling pathways begins when a Wnt ligand binds to its receptors at the cell membrane. However, the question of how a Wnt moves from a producing cell to fulfill its paracrine function in a tissue remains highly debated[10]. The transport of signal components such as ligands and receptors can be facilitated by signaling filopodia known as cytonemes[11–13]. Indeed, recent high-resolution imaging experiments in zebrafish demonstrated that specialized cytonemes are fundamental in Wnt trafficking in vertebrates[14,15]. The regulated generation and function of cytonemes is critical as it impacts directly on the signaling range and signaling gradient. Specifically, the number and length of cytonemes generated by a Wnt source cell influence events, such as zebrafish neural plate patterning during embryogenesis[14]. However, the molecular mechanism regulating these attributes during zebrafish gastrulation to allow neural plate patterning is still unclear.

Unlike in flies, Wnt-dependent activation of the receptor-tyrosine kinase-like orphan receptor 2 (Ror2) is thought to act as a crucial receptor of the Wnt/PCP pathway[16], and in turn drives de novo biogenesis of Wnt8a-positive cytonemes[17]. The subsequent formation of cytonemes is influenced by activation of cytoskeletal regulators, such as the small Rho GTPase Cdc42, which controls actin polymerization[14,18]. Ror2 is thought to function by forming a protein complex with other PCP regulatory proteins at the plasma membrane, at a sub-membrane region, or at cell–cell junctions. This PCP core complex can include the key component, the four-pass transmembrane protein, Van-Gogh-like (Vangl) 1/2 in vertebrates. As Vangl2 lacks any known receptor or enzymatic activity[19], protein–protein interaction domains of Vangl2 are likely to modulate downstream signaling. In mouse, Wnt ligands bind to both Ror2, which recruit CK1δ/ε through Dvl. Subsequently, CK1δ/ε activates Vangl2 by phosphorylation[20,21], similarly CK1 phosphorylates Vangl2/strabismus in Drosophila[22]. Supporting a functional role for

Vangl2, its knockdown inhibits axon outgrowth by inhibition of filopodia formation[23]. It has been suggested that this downstream signaling pathway is triggered in a Wnt concentration-dependent manner in mouse limb development[21]. During zebrafish gastrulation, Vangl2 is asymmetrically localized at the plasma membrane and localizes to forming protrusive membranes[24,25]. It is currently unclear if the Vangl2-positive filopodia perform signaling functions in tissue organization.

Here we show that Vangl2—together with Wnt8a and Ror2—is loaded on cytoneme tips. Vangl2 positive cytoneme tip complex activates JNK signaling to increase cytoneme length and the number of cytoneme contacts. We find that Vangl2 function during cytoneme emergence is vital for paracrine Wnt/β-catenin signaling in both human tissue culture and zebrafish embryo. Concordantly, blockage of Vangl2 function or JNK signaling leads to quickly collapsing signaling filopodia. Consequently, impairment leads to a reduction of both Wnt dissemination and paracrine Wnt signaling in human cancer tissue culture, the zebrafish embryo and the mouse intestinal crypt. Based on our findings, we developed a mathematical model of how changes in cytoneme length and contacts would affect embryogenesis. This mathematical model of morphogen distribution in the zebrafish gastrula predicts that increased Wnt cytoneme length and number of contacts leads to extended signaling range, and to altered patterning with fuzzy compartment boundaries. We confirm these predictions in vivo during zebrafish neural plate patterning. These findings suggest that the activity of the Vangl2-positive PCP complex determines the emergence of Wnt-positive cytonemes during development and tissue homeostasis in vertebrates.

## Results

**Vangl2 together with Wnt8a and Ror2 form the cytoneme tip complex.** The PCP signaling component Ror2 is an essential regulator of Wnt8a-positive cytoneme emergence[17]. We investigated the interconnected role of additional PCP family members on cytoneme regulation. The four-transmembrane PCP protein Vangl2 is a PCP core member that is localized at the tips of filopodia extending from both neurons, as well as in forming membrane protrusions in gastrula cells during zebrafish gastrulation[23,25,26]. Upon Wnt activation, Ror2 recruits CK1δ/ε, which phosphorylates Vangl2 to activate downstream signaling, such as c-Jun N-terminal kinase (JNK) signaling[16,20,21,27]. To analyse the localization and function of PCP components together with Wnt8a, we first sought to establish an in vitro test system to monitor cytoneme behavior using zebrafish PAC2 fibroblast cells in culture. First, we asked if PAC2 cells transported endogenous Wnt8a by filopodial protrusions. We found that these fibroblasts have numerous, dynamic filopodial protrusions and that endogenous Wnt8a can be detected on these filopodia, similar to over-expressed fluorescently-tagged Wnt8a (Fig. 1A, B and Supplementary Fig. 1A, B). Therefore, we define filopodial protrusions as Wnt8a-bearing cytonemes when they co-localized with fluorescently-tagged Wnt8a along the filopodium or at the filopodium tip (Fig. 1A, B). Next, we asked if filopodia can also be decorated with Vangl2 protein. As there is no suitable antibody for immunohistochemistry against zebrafish Vangl2 available, we stained the Wnt signaling active gastric cancer cells AGS with a human anti-Vangl2 antibody and, indeed, could find that endogenous hVangl2 is also localized to the tip of filopodia of AGS cells (Supplementary data Fig. 1C). We measured fluorescent intensity along the PAC2 cytonemes (Fig. 1G) and found that glycophosphatidylinositol (GPI)-anchored, membrane-bound mCherry (mem-mCherry) was evenly distributed along the length of cytonemes from tip to base, whilst Wnt8a-GFP was

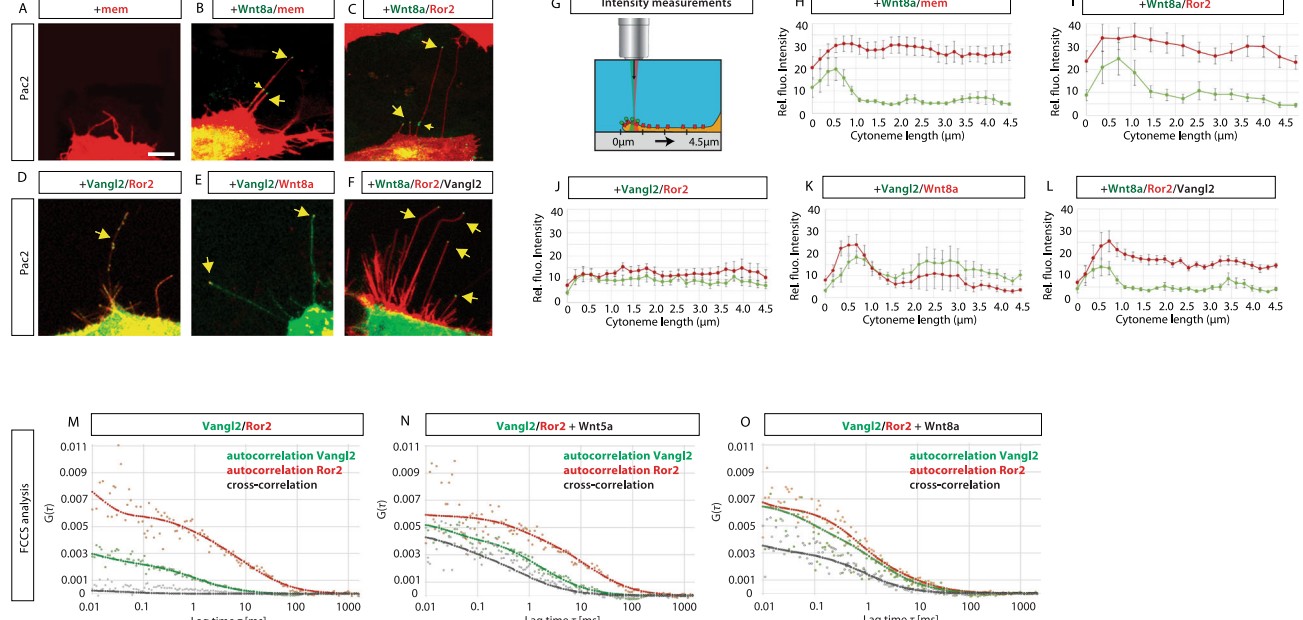

**Fig. 1 Vangl2 is present on the tips of Wnt8a positive cytoneme.** A–F PAC2 zebrafish fibroblasts transfected with indicated constructs and analysed live at 24 h of post-transfection. Yellow arrows indicate Wnt8a on tips of cytonemes. White scale bar equals 5 μm. ($n = 25, 9, 14, 6, 25,$ and 14 cells). G Fluorescent intensity measurements were recorded (**H–L**) along cytoneme length as illustrated in **G**, from tip at 0–4.5 μm along cytoneme. Cytonemes fluorescent intensity measurements were taken up to 4.5 μm from the cytoneme tip in each case. **H–L** Relative fluorescent intensity analysis (gray value) of tagged proteins Wnt8a-GFP, Wnt8a-mCherry, Mem-mCherry, Ror2-mCherry, and GFP-Vangl2, relative pixel intensity values were measured along cytoneme length starting at the cytoneme tip, ($n = 17, 12, 12, 14, 11,$ and 12 filopodia). Standard error of the mean (SEM) = 1. **M–O** For the fluorescence correlation spectroscopy (FCS) analysis, a focused laser spot was scanned across the membrane for 16s while the intensity was measured as a function of time ($G(t)$). Auto-correlation curve in red and green for Ror2 and Vangl2, respectively. Cross-correlation curve in gray. Fluorescent cross-correlation spectroscopy (FCCS) revealed cross-correlation of Ror2-mCherry and GFP-Vangl2 when exposed to Wnt5a protein or Wnt8a protein. ($n = 5$ measurements/condition). CTRL $D_2$ (Ror2) = $(37 \pm 5)$ μm$^2$ s$^{-1}$, $D_2$ (Vangl2) = $(23 \pm 4)$ μm$^2$ s$^{-1}$, $K_D > 1100$ nM; Wnt5a $D_2$ (Ror2) = $(14 \pm 3)$ μm$^2$ s$^{-1}$, $D_2$ (Vangl2) = $(19 \pm 5)$ μm$^2$ s$^{-1}$, $K_D = 173$ nM; Wnt8a: $D_2$ (Ror2) = $(11 \pm 2)$ μm$^2$ s$^{-1}$, $D_2$ (Vangl2) = $(16 \pm 3)$ μm$^2$ s$^{-1}$, $K_D = 138$ nM. Source data are provided as a Source Data file.

predominantly localized to the tip of the cytoneme (Fig. 1B, H). Next, we transfected plasmids encoding Ror2-mCherry and GFP-Vangl2 to investigate the localization of these PCP proteins in the PAC2 cells. Ectopically expressed Ror2-mCherry was membrane localized (Fig. 1C) with a slight increase in fluorescence of Wnt8a intensity at the tip (Fig. 1L). GFP-Vangl2 and Ror2-mCherry expression was membrane localized and present all along the cytonemes (Fig. 1D, J), whereas GFP-Vangl2 can be also seen at the filopodia tip of PAC2 cells (Supplementary Fig 1D). GFP-Vangl2 accumulated dynamically together with Wnt8a at cytoneme tips (Fig. 1E) with an increase in fluorescence intensity at the tip (Fig. 1K). We find that in the presence of over-expressed untagged Vangl2, Ror2-mCherry accumulated more strongly at the cytoneme tip (Fig. 1F, L).

To further characterize the interactions between Wnt8a, Vangl2, and Ror2, we used fluorescence cross-correlation spectroscopy (FCCS) to determine the formation of a receptor complex[17]. We performed an FCCS analysis at the plasma membrane of PAC2 cells expressing GFP-Vangl2 and Ror2-mCherry (Fig. 1M). We found the cross-correlation of Vangl2 and Ror2-mCherry if treated with Wnt5a compared to the untreated control in accordance with previous published biochemical data in mouse[20,21] (Fig. 1N). Similarly, we observed cross-correlation of Vangl2 and Ror2, when PAC2 cells were treated with Wnt8a protein (Fig. 1O). We conclude that Wnt8a can induce the formation of a Ror2/Vangl2 complex. This shows that Wnt8a induces a Vangl2/Ror2 complex at the tip of the cytoneme, suggesting a potential role in emergence and function of Wnt cytoneme.

**Vangl2 influences cytoneme formation in zebrafish fibroblasts**. To characterize the function of Vangl2 in cytoneme formation, we analysed the effect of Vangl2 on cytoneme number and length in vitro. PAC2 fibroblasts have numerous, dynamic filopodial protrusions, which are positive for Wnt8a. We found on average 5.7 cytonemes per cell in control cells with an average length of 7.9 μm (Fig. 2A, K, L and Supplementary Fig. 2A, B). The average cytoneme length was slightly longer than Wnt8a-negative filopodia lengths (Supplementary Figs. 3 and 4). We found that 23% of the observed cytonemes were 10 μm or longer (Fig. 2M). Activation of Vangl2 caused the average cytoneme length to increase significantly by 187.3% (Fig. 2C, L, M and Supplementary Fig. 2B). While in control cells only 23% of cytonemes were >10 μm, in the Vangl2-overexpressing cells 68.9% were >10 μm, and 43.8% were >20 μm (Fig. 2M). Ror2/Vangl2 double-transfected cells had a similar and significant increase in Wnt8a positive cytoneme length (Fig. 2D, L and Supplementary Fig. 2B), and in addition displayed more branching (Fig. 2D). Vangl2 lacking the phosphorylation sites exhibit a dominant negative effect on signaling[21]. So we used an N-terminal deletion mutant of Vangl2 (ΔN-Vangl2) lacking the two Ser/Thr phosphorylation clusters and the all-phospho Vangl2 mutant S5–17A::S76–84A, in which 10 Ser/Thr are replaced by Ala (Vangl2[10A]) to block Vangl2-mediated signaling[21]. Ectopic expression of the Vangl2[10A] led to a significant reduction in the number of cytonemes and ΔN-Vangl2 and Vangl2[10A] to a reduction in the number of long cytonemes, and an increase in the number of short cytonemes (Fig. 2E, G, K, M and Supplementary Fig. 2A, B). Vangl2[10A] also showed significant reduction in all filopodia numbers

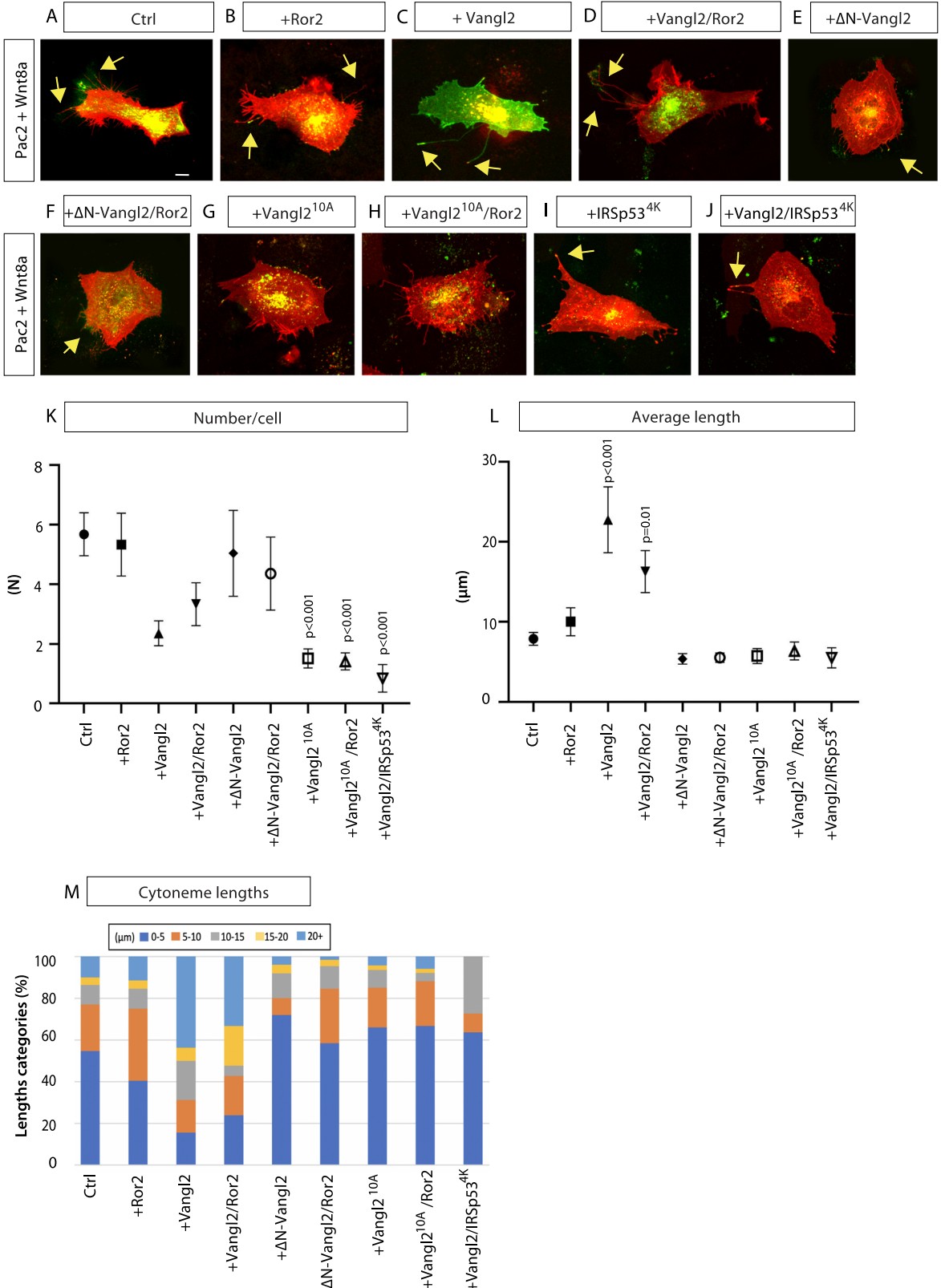

**Fig. 2 Vangl2 controls the emergence of Wnt8a cytonemes in fibroblasts. A–J** Wnt8a-positive PAC2 zebrafish fibroblasts transfected with indicated constructs, imaged and analysed live 24 h of post-transfection. Yellow arrows indicate examples of Wnt8a positive cytonemes. Scale bar = 10 μm. **K** Number of Wnt8a positive cytonemes per cell (*n* = number). (*n* per condition = 25, 9, 14, 6, 25, 14, 31, 36, 13 cells). **L** Length of Wnt8a positive cytonemes in PAC2 cells (μm). (*n* per condition = 139, 52, 32, 21, 131, 65, 47, 51, 11 cytonemes). **M** Breakdown of the percentage of Wnt8a positive cytoneme lengths into 0–5, 5–10, 10–15, 20+ μm categories. Graphs represent mean and standard error of the mean. **K**, **L** Two-sided Kruskal–Wallis tests with Bonferroni correction for multiple tests. Statistical significance: *p* ≤ 0.05. SEM = 1. Corresponding dot plots are shown in Supplementary Fig. 2, and analysis of PAC2 filopodia is shown in Supplementary Figs. 3 and 4. Source data are provided as a Source Data file.

and ΔN-Vangl2 and Vangl2[10A] showed reduction in all filopodia length compared to control (Supplementary Figs. 3 and 4). Co-expression of Ror2 did not reverse the phenotype (Fig. 2F, H, K, L, M), suggesting that Vangl2 functions downstream of Ror2.

The I-Bar protein insulin receptor tyrosine kinase substrate p53 (IRSp53) plays an essential role in filopodia formation, and connects filopodia initiation and maintenance by assembling the actin scaffold[28]. IRSp53 is localized to the tips of filopodia and overexpression leads to the de novo formation of filopodia[29]. IRSp53[4K] has four lysine residues in the actin-binding sites mutated to glutamic acid, which strongly reduce filopodia formation[17,30,31]. Overexpression of IRSp53[4K] reduced the number and decreased the length of cytonemes (Fig. 2I), and Vangl2 activation could not compensate for the loss of IRSp53 function (Fig. 2J–M and Supplementary Figs. 2 and 3J, K), suggesting that Vangl2-mediated induction of cytonemes operates upstream of the filopodia nucleation machinery.

**Vangl2 controls the length of Wnt8a cytonemes during zebrafish gastrulation.** We next investigated the role of Vangl2 in cytoneme regulation in vivo in zebrafish embryos. First, we asked if Vangl2 can be localized to cytonemes of embryonic cells during zebrafish gastrulation. To do this, we analysed the localization of Vangl2 using a transgenic line expressing a zebrafish GFP-Vangl2 N-terminal fusion protein under the control of the Vangl2 promoter[32]. In *Tg(vangl2:GFP-Vangl2)* embryos injected with mRNA of *mem-mCherry*, we found that epiblast cells have dynamic filopodial protrusions and that GFP-Vangl2 can be detected on these filopodia with an accumulation at the tips (Fig. 3A, A' and Supplementary movie 1A, 1B). Overexpression of Ror2 leads to the formation of more cytonemes in vivo[17] and Vangl2 is similarly localized to the cytonemes of Ror2-positive cells (Fig. 3B, B' and Supplementary movie 2A, 2B). Next, we generated clones of cells in intact zebrafish embryos by micro-injecting mRNA of *mem-mCherry* and *wnt8a-GFP*, together with either *vangl2*, *ΔN-vangl2*, the all-phospho mutant *vangl2[10A]*, and *IRSp53[4K]* mRNA at the 8–16-cell stage. At 5hpf (50% epiboly), we imaged individual clones of cells in the zebrafish gastrula (Fig. 3C–G). In particular, we focused on visualization of Wnt8a-positive cytonemes within the embryo. We found that, like what occurred in cultured PAC2 cells, expression of *vangl2* mRNA led to significantly longer cytonemes per cell compared to control (Fig. 3C, D, H, I and Supplementary Fig. 5A, B). The average length of cytonemes significantly increased by 39.6% (Fig. 3I, J). The cytonemes in Vangl2-expressing embryonic cells were likewise found to branch abnormally, form multiple contact points and extend over larger areas between cells compared to control (Fig. 3D, Supplementary Fig. 5C, and Supplementary movie 3, 4). Expression of *ΔN-vangl2* or *vangl2[10A]* did not significantly change the length of cytonemes compared to control however, it significantly reduced the number (Fig. 3E, F, H, I and Supplementary movie 5), suggesting that the observed lengthening and cytoneme induction requires Vangl2 function. Next, we used the IRSp53[4K] mutant to block filopodia formation. Co-expression of IRSp53[4K] and *vangl2* led to a significant reduction of filopodia including Wnt8a-positive cytonemes, and at the same time cells starts to form more tubular structures (Fig. 3G)—similar to our observation in PAC2 fibroblasts (Fig. 2I, J and Supplementary movie 6).

Next, we asked if the increased average length of cytonemes upon Vangl2 activation leads also to an increase of contact sites. Therefore, we quantified the number of contact sites of one cytoneme on the receiving cells, and found that a cytoneme on average contacts the neighboring cell only once (Fig. 3K–M and Supplementary movie 3). However, we found that this number

doubles in Vangl2 expressing cells, and in some cases, we counted over five contact sites on various cells by a single cytoneme (Fig. 3K–M and Supplementary Movie 4). This effect could not be observed after overexpression of *ΔN-vangl2* or the *vangl2[10A]* mutant (Fig. 3M, Supplementary movie 5). Furthermore, the effect of Vangl2 overexpression can be reversed by blockage filopodia by reducing IRSp53 function (Fig. 3M). Therefore, we conclude that Vangl2 is necessary but not sufficient for cytoneme induction, and it regulates both Wnt8a cytoneme length and cell contact sites—both in vitro and in vivo.

**Vangl2-mediated JNK activation is required to stabilize cytonemes.** We next wanted to investigate the mechanisms by which Vangl2 regulates cytoneme stability. Wnt/PCP can transduce signals through the receptors Frizzled and Ror2 to Rac-JNK and RhoA-ROCK signaling cascades in a context-dependent manner[33]. Apart from its nuclear functions, JNK also directly regulates the cytoskeleton by phosphorylation of diverse cytoplasmic targets including proteins directly interacting with the actin cytoskeleton, as well as microtubule-associated proteins[34]. Similarly, the RhoA-ROCK signaling cascade can induce actin cytoskeletal reorganization and cell movement[7]. In addition, there is crosstalk between these pathways; RhoA can activate JNK during convergent extension movement in Xenopus, and loss of RhoA can be rescued by over-expression of JNK1[35].

We recently reported results from a cell-culture-based screen to identify kinases that regulate cytoneme formation[17]. In this screen, in addition to Ror2, we identified several key family members of the JNK signaling pathway including MKK4 and JNK3 as positive regulators of Wnt8a cytoneme length. After receiving external signals, MAP kinase kinases (MKK) phosphorylate and activate c-Jun N-terminal kinases (JNK). In turn, the JNKs phosphorylate a number of transcription factors, primarily components of AP-1[36]. In our screen, forced expression of MKK4 led to an increase in the average length of cytonemes by 36.4%, whereas JNK3 expression resulted in an increase of 19.7%. To further probe the involvement of JNK signaling, we carried out a reporter assay in HEK293T cells, which express a very low level of endogenous Wnt ligands and a defined set of Fzd receptors[37]. In cells with low JNK activity, the KTR-mCherry reporter[38,39], is localized to the nucleus (Supplementary Fig. 6A). However, upon activation of JNK signaling, phosphorylation of the JNK-KTR-mCherry reporter causes it to translocate to the cytoplasm[38]. Control HEK293T cells have low JNK activity, shown by nuclear localization of KTR-mCherry (Fig. 4A, B). Transfection of Ror2, Vangl2 or a combination of both did not significantly change the ratio of cytoplasmic to nuclear signal (Fig. 4A, B). Remarkably, addition of either Wnt5a or Wnt8a protein with Ror2 and Vangl2 led to a significant increase in JNK activity in Ror2/Vangl2-expressing cells, as seen as movement of KTR-mCherry from nucleus to cytoplasm (Fig. 4A, B). This suggests that Wnt protein with Ror2 and Vangl2 initiates JNK signaling in HEK293T cells. Wnt8a did not activate JNK signaling in cells transfected with ΔN-Vangl2 (Fig. 4B). This indicates that Vangl2 is a key downstream element in Ror2-Wnt activated JNK signaling in HEK293T cells.

We observed that Wnt8a cytonemes form and retract within some tens of minutes (Supplementary Movie 7)[14]. To assess the role of JNK signaling in this process, we treated Wnt8a-GFP/mem-mCherry transfected PAC2 cells with a small molecule inhibitor of JNK kinase activity. In vertebrates SP600125 specifically inhibits all three JNKs (JNK1-3) within minutes without inhibition of ERK1 or ERK2, phospho-p38, or ATF2[40]. We then recorded and analysed the effect of JNK

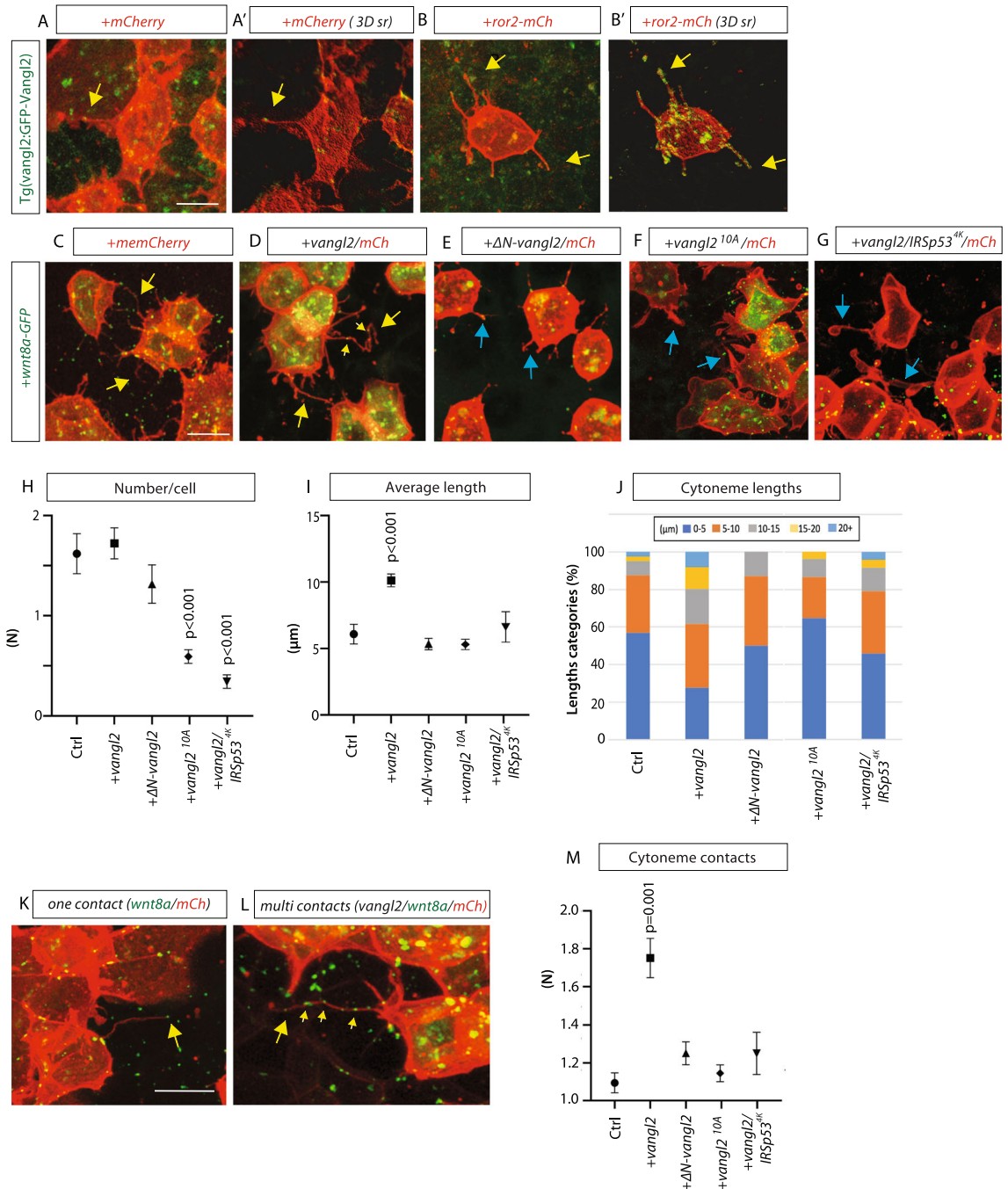

**Fig. 3 Vangl2 regulates length and contacts of Wnt8a cytonemes in the zebrafish embryo. A**, **B'** *Tg(vangl2:GFP-Vangl2)* zebrafish embryos were injected with mem-mCherry (mCh) or Ror2-Mcherry. **A'**, **B'** 3D renders (3D sr) of A&B at 5hpf. Yellow arrows show examples of Vangl2 positive cytonemes. **A–A'** *n* = 6 embryos. **B–B'**: *n* = 8 embryos. **C–G** Wild type zebrafish embryos were injected with indicated constructs and imaged at 5hpf. Yellow arrows show examples of Wnt8a positive cytonemes. Blue arrows show examples of thick membrane protrusions. Scale bar = 10 μm. **H** Number of Wnt8a positive cytonemes per cell (*N* = number). (*n* per condition = 3, 6, 3, 9, 7 embryos, *n* = 87, 123, 44, 154, 93 cells). **I** Length of Wnt8a positive cytonemes (μm). (*n* per condition = 81, 269, 54, 82, 24 cytonemes). **J** Breakdown of the percentage of Wnt8a positive cytoneme lengths into 0–5, 5–10, 10–15, 20+ μm categories. **K** Wnt8a/Mem-mCh cytoneme with one wnt8a + ve contact point. Scale bar = 10 μm. **L** Vangl2/Wnt8a/mem-mCh cytoneme with multiple contact points **M** Quantification of number of Wnt8a-positive contact points per cytoneme. Wnt8a positive points were counted when at tips or at a fixed junction or "turning point" (*n* = 32, 125, 80, 81, 24 cytonemes). Graphs represent mean and standard error of the mean. Corresponding dot plots are shown in Supplementary Fig. 5. **H**, **I**, **K** Two-sided Kruskal–Wallis tests with Bonferroni correction for multiple tests. Statistical significance: *p* ≤ 0.05. SEM = 1. Source data are provided as a Source Data file.

blockage on protrusion length and number (Fig. 4C–E). We found that Wnt cytonemes collapse and retracted following the addition of SP600125 (Fig. 4C), whereas cytonemes in DMSO-treated PAC2 cells are unchanged. A time course revealed that over the course of 120 min, control cells showed no significant change in the number or length of signaling filopodia (Fig. 4D, E and Supplementary Fig. 6B). However, JNK inhibition caused a significant reduction in average relative protrusion length and

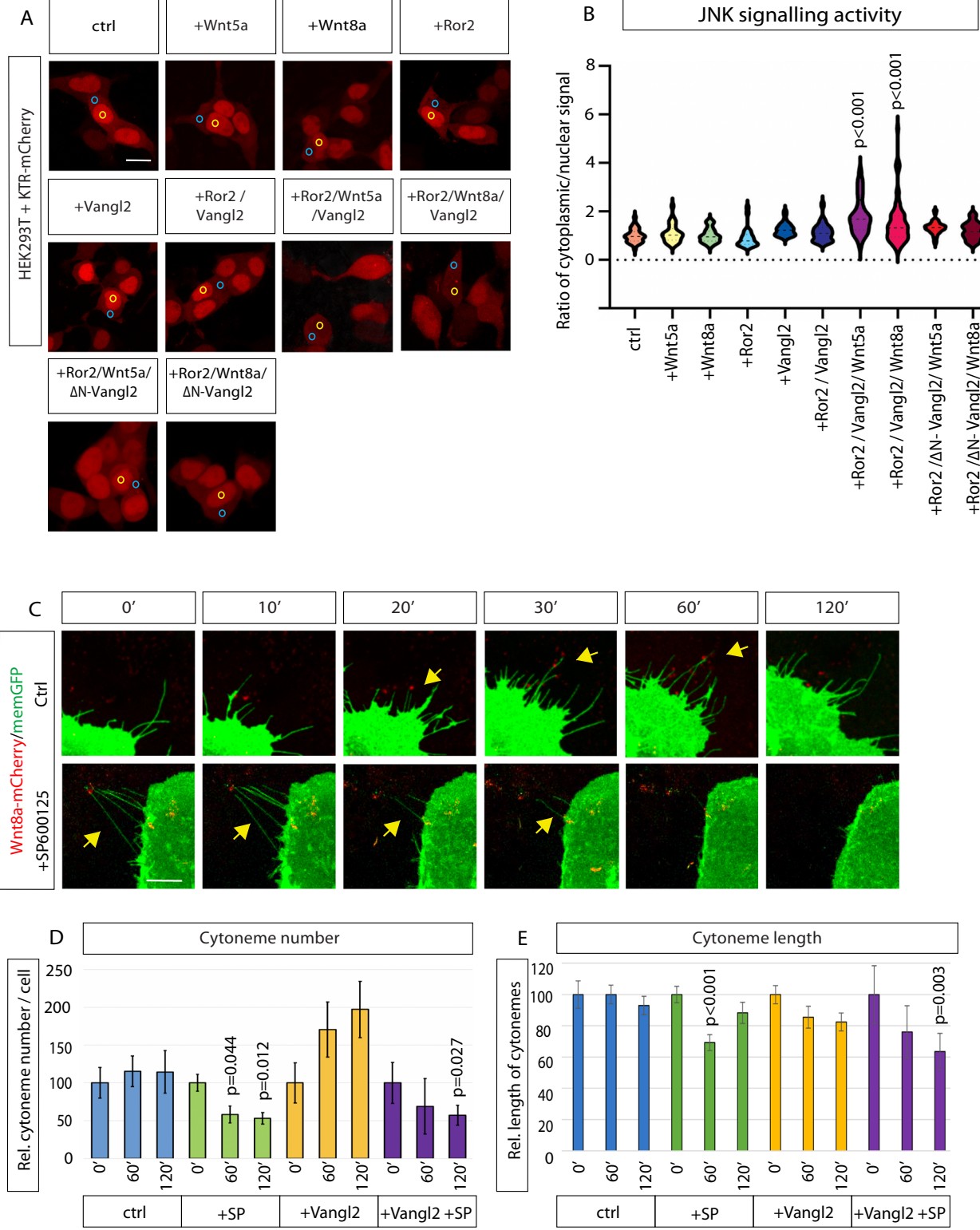

number after 1 h. Vangl2-expressing cells, which had longer protrusions (as in Fig. 2L) prior to the addition of SP600125, also had a significant change in the number and length of cytonemes compared to untreated Vangl2-expressing fibroblasts. A detailed analysis of cytoneme lengths showed a specific loss of extremely long signaling filopodia as a result of JNK inhibition (Supplementary Fig. 6B). We also tested the involvement of RhoA/ROCK signaling as it can be similarly activated downstream of PCP signaling to regulate cytoskeletal

re-arrangement. To do so, we treated cytoneme-bearing cells with Y-27632, an antagonist of Rho-associated kinase (RhoA/ROCK)[41]. While JNK inhibition had a significant effect at 60 min, Y-27632 treatment produced only a modest and non-significant reduction of protrusion length after 5 h (Supplementary Fig. 6C–G). We conclude that JNK signaling is the primary signaling cascade required for fast cytoneme formation downstream of Wnt/Vangl2/Ror2 signaling and it contributes to the formation of long Vangl2-positive cytonemes.

**Fig. 4 Vangl2 activates JNK signaling to control the formation of Wnt8a cytonemes. A** HEK293T cells all transfected with KTR-mCherry and Wnt5a or Wnt8a protein, plus Ror2; Vangl2; Ror2 & Vangl2; Ror2 & ΔN-Vangl2. Wnt8a protein and Wnt5a protein (500 ng/ml) was also added to some conditions 24 h prior to imaging. Blue circle = nuclear, yellow circle = cytoplasmic. **B** Violin plot of normalized ratio of cytoplasmic to nuclear signal of KTR-mCherry. Increased ratio indicates increasing JNK activity. ($n = 69, 32, 32, 25, 30, 24, 43, 75, 60, 49$ cells). Two-sided Kruskal–Wallis tests with Bonferroni correction. SEM = 1. **C** Time series of PAC2 cells at 0 (0′), 10, 20, 30, 60, and 120 min in control cells and cells post 20 μM SP600125 treatment. **D** Relative number of filopodia per cell in relation to time = 0 h, at 0, 60, 120 min. ($n = 3, 6, 10, 4$ cells. $n$ = filopodia at 0, 1, 2 h = (102/111/109, 251/156/122, 158/185/215, 58/50/44). $p$ values over + Sp600125 60′ & 120′ (green bars) significant to 0′. $p$ value over Vangl2 + SP600125 120′ (purple) significant to Vangl2 0′ (yellow). **E** Relative filopodia length (μm) in relation to time = 0 h, at 0, 60, 120 min. ($n = 3, 6, 13, 4$ cells. $n$ = filopodia at 0, 1, 2 h = (102/111/109, 251/156/122, 232/274/295, 58/50/44). $p$ value over + Sp600125 60′ (green) significant to 0′. $p$ values over Vangl2 + SP600125 120′ (purple) significant to Vangl2 0′ (yellow). Corresponding dot plot for **D**, **E** shown in Supplementary Fig. 6B. Statistical significance: $p \leq 0.05$. **D**, **E** Two-sided Kruskal–Wallis tests without Bonferroni correction. SEM = 1. Scale bar = 10 μm. Source data are provided as a Source Data file.

**Vangl2-controlled cytonemes regulate Wnt/β-catenin signaling in neighboring cells.** Next, we investigated the effect of Vangl2 on cytoneme-mediated Wnt protein delivery by investigating Wnt/β-catenin signal activation in the receiving cells (Fig. 5). To measure paracrine signal activation, we used a sensitive reporter system of AGS gastric cancer cells—which are primed for Wnt/β-catenin signaling—transiently transfected with a Wnt/β-catenin reporter with seven TCF-responsive elements driving expression of nuclear mCherry (7xTCF-nls-mCherry[42]). AGS cells have also been shown to exhibit Ror2-dependent Wnt8a cytonemes[17]. We co-cultivated AGS cells transiently expressing combinations of Wnt8a, Vangl2, Ror2, and ΔN-Vangl2 together with the STF-mCherry reporter cells (Fig. 5A, B). The expression of nuclear mCherry significantly increased in the receiving cells when Wnt8a was expressed in the source cells (Fig. 5Bii, C). This suggests that Wnt8a can activate the Wnt/β-catenin pathway in a paracrine way. Expression of Vangl2 alone in the source cells did not induce a Wnt/β-catenin response in the receiving cells (Fig. 5Biii, C). However, Wnt8a/Vangl2 co-transfection, which we showed in zebrafish cells increased cytoneme contact points and length, could significantly increase STF reporter activation in receiving cells in comparison to control and Wnt8a only (Fig. 5Biv, C). The significant increase of Wnt signal transmission to the responding cells is dependent on Vangl2, because co-transfection of Wnt8a/ΔN-Vangl2 showed no detectable effect on reporter activation compared to Wnt8a expressing source cells (Fig. 5Bv, C). Next, we tested if the Vangl2-dependent increase of paracrine Wnt/β-catenin signal activation depends on Ror2 function. We found that expression of Wnt8a/Ror2/Vangl2 in the producing cells further increased reporter activation in the receiving cells (Fig. 5Bvi, C), and this was abrogated by co-expression with ΔN-Vangl2 or kinase-dead Ror2 (Ror2[3i]) (Fig. 5Bvii,Bviii,C). These data indicate that the activity of Wnt8a in source cells is markedly enhanced by two factors known to increase cytoneme number and length, consistent with the transmission of the Wnt8a signal by cytonemes. At this point, we could not exclude that formation of longer cytonemes and increased activation of the Wnt8a signaling in neighboring cells by activation of Vangl2 in the source cells are two independent events, acting in parallel. Therefore, we blocked cytoneme formation in cells expressing Wnt8a and Vangl2 by co-expression of the mutant form of ISRsp53[4K] (Fig. 5D, E). We find that blockage of filopodia formation in these cells reduces significantly the paracrine signaling activity by Vangl2/Wnt8a expressing cells. The signaling capacity of Wnt8a/Vangl2/IRSp53[4K] is similar to the control cells (Fig. 5D, E), suggesting that the Vangl2-regulated Wnt8a transport requires mainly cytonemes.

Stem cells are highly dependent on extrinsic cues derived from their microenvironment; however, it is still unclear how signaling is controlled within such a niche environment. Wnt/β-catenin signals are an essential component of a wide range of stem-cell niches, including that of the gastrointestinal epithelium[43]. The intestinal stem-cell niche is regulated by both Wnts and RSPO3,

which are supplied predominantly by stromal myofibroblasts[44,45]. To test for the requirement for cytoneme-mediated Wnt transport in stem cell regulation, we investigated the influence of VANGL on WNT cytonemes in the mouse intestinal crypt. In particular, the stem cells at the bottom of intestinal crypt require constant Wnt/β-catenin signaling for tissue homeostasis[46–49]. In vivo, subepithelial myofibroblasts, described as PDGFRα + telocytes, are the major source of physiologically relevant WNTs to maintain these intestinal crypts[44,50,51]. We have shown the ability of these myofibroblasts to provide essential WNTs is compromised when they lack cytonemes as a result of siRNA mediated ROR2 inhibition[17]. We used crypt organoids as an optimal system to test our hypothesis that telocytes also require VANGL to generate cytonemes to distribute WNT proteins in the mouse intestinal crypt. VANGL1 and VANGL2 expression in telocytes was reduced by siRNA-mediated knockdown (Supplementary Fig. 7A). The number of filopodia was significantly reduced after *Vangl1* knockdown and even more markedly reduced after double knockdown of *Vangl1/Vangl2* (Fig. 6A, B). In parallel, we found a reduction in filopodia length in all three knockdown experiments with the most significant reduction after double knockdown of *Vangl1/Vangl2* (Fig. 6C). Next, we used an organoid formation assay to analyse the requirement for VANGL-dependent WNT cytonemes (Fig. 6D). Organoids of WNT-deficient *Porcn*[−/−] crypt cells need to be co-cultivated with WT Wnt-producing telocytes for maintenance. Telocytes interact with crypt organoids and supply them with signaling factors such as WNT proteins. Notably, we were not able to observe long filopodia in telocytes upon knockdown of *Vangl1/Vangl2* or *IRSp53* (Fig. 6A–D). After simultaneous knockdown of *Vangl1* and *Vangl2* in the WNT-producing telocytes, we further observed a strong decrease in the number of organoids (Fig. 6E). In addition, knockdown of *IRSp53* led to a similar reduction in organoid counts (Fig. 6F). This suggests that the WNT cytonemes from the telocytes are required for the induction and maintenance of the intestinal crypt organoids and that VANGL1/VANGL2 are crucial for their formation.

Simulation predicts an important role for Vangl2-controlled cytonemes in the zebrafish gastrula.

On the basis of these findings, we hypothesized that Vangl2 function in the Wnt source cells is crucial for Wnt dissemination via cytonemes which we examined in silico. To quantitatively test the consequences of altered Vangl2 function on gradient formation and tissue patterning in the zebrafish neural plate, we created an agent-based simulation of morphogen distribution via cytonemes using the Chaste modeling software[52,53]. First, we generated a 2D model of the zebrafish gastrula based on the positional information of every cell during the first 10 h of zebrafish gastrulation[54,55], representing a portion of the overall gastrula. We defined the population of marginal cells as Wnt8a source cells and the overlying epiblast cells as Wnt-receiving neural plate cells (Fig. 7A, B). The simulation takes into account

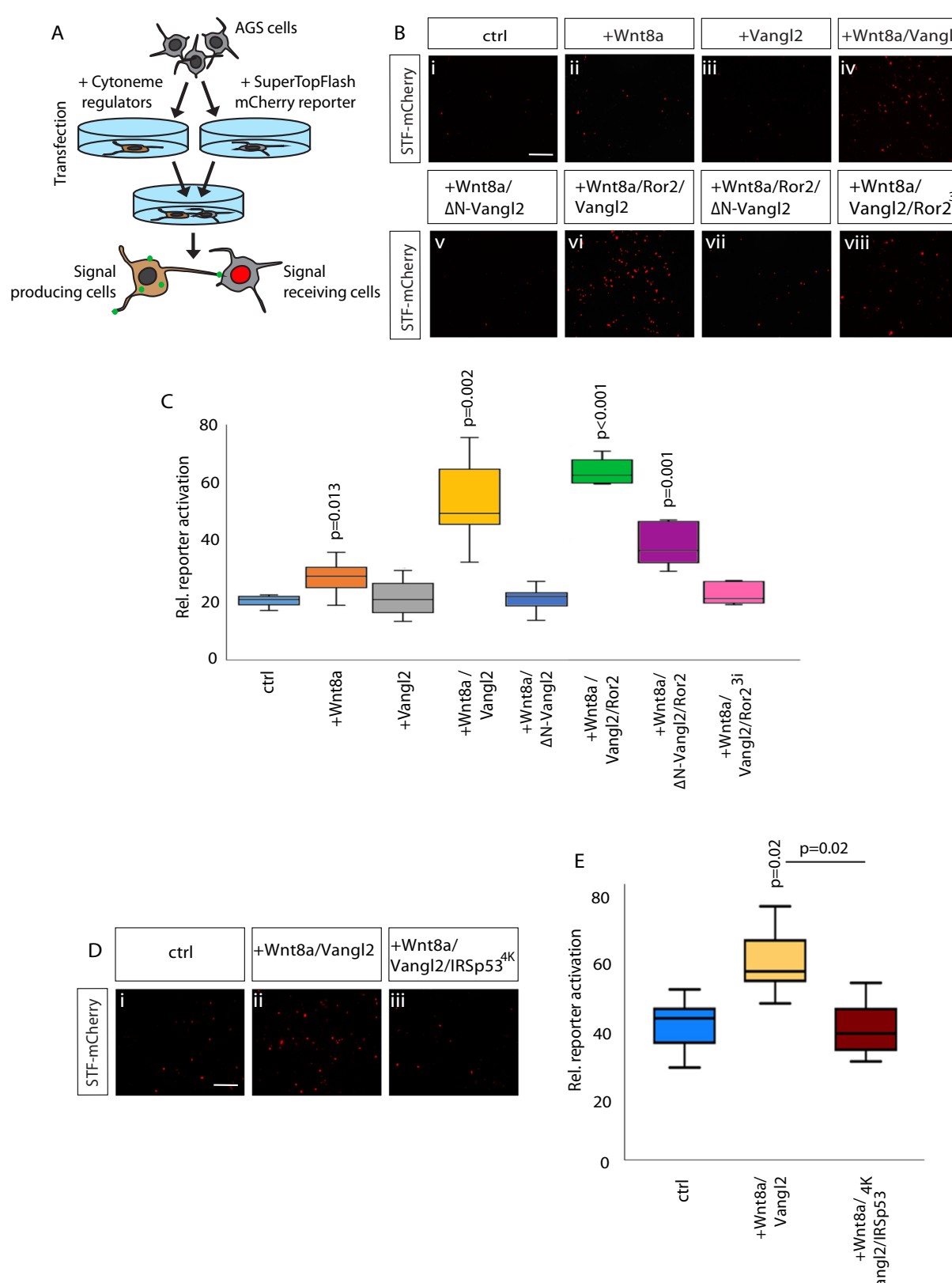

ligand transport by cytonemes, ligand decay and the migration and proliferation of epiblast cells using the agent-based simulation approach (Fig. 7C–F). We employed cytonemes as the exclusive transport mechanism from the producing marginal source cell group to the target cell group. We made the assumption that all cells are initially at the animal pole then

migrate and intercalate to produce a thin tissue, which covers the yolk during the epiboly movement and Wnt transport process. We found that cytonemes can distribute Wnt8a in a graded manner in the dynamically evolving target tissue of the zebrafish embryo during gastrulation. Cells receiving a high concentration of Wnt8a acquire hindbrain fate, according to a pre-defined Wnt

**Fig. 5 Vangl2 activity in the Wnt source cells regulates paracrine Wnt/β-catenin signal activation. A** Schematic of SuperTOP Flash (STF) reporter co-cultivation assay in AGS cells. AGS cells transfected with STF reporter were co-cultivated AGS cells transfected with combinations of with Wnt8a, Vangl2, Ror2 and ΔN-Vangl2 plasmid. **B** AGS cell STF reporter activation at each condition (i–viii). Scale bar = 10 μm. **C** Relative STF reporter activation in cells when co-cultured with control; Wnt8a; Vangl2; Wnt8a/Vangl2; Wnt8a/ΔN-Vangl2, Wnt8a/Ror2/Vangl2; Wnt8a/Ror2/ΔN-Vangl2 and Wnt8a/Vangl2/Ror2[3l]. ($n = 5, 10, 5, 5, 5, 5, 5, 5$ repeats). **D** AGS cell STF reporter activation after IRSp53[4k] addition: (i–iii); control; Wnt8a/Vangl2; Wnt8a/Vangl2/IRSp53[4k]. Scale bar = 10 μm. **E** Relative STF reporter activation in cells when co-cultured with control; Wnt8a; Vangl2; Wnt8a/Vangl2; Wnt8a/Vangl2/IRSp53[4k]. ($n = 20$ biological repeats across two independent experiments). **C, E** Data represented as box and whisker plots. Whiskers define the minimum and maximum values. Bounds of box indicate the 25th and 75th percentile, center line indicates the median, **C** calculated with exclusive median. **C** Student's *t*-test and **E** One-way ANOVA test plus Tukey's post-hoc test. Statistical significance: $p \leq 0.05$. Source data are provided as a Source Data file.

threshold, whereas, cells receiving a lower concentration acquire forebrain/midbrain fate. We further find the formation of a stable boundary between midbrain and hindbrain (MHB) due to a sharp drop of the morphogen concentration across the boundary. Next, we tested two scenarios with varying ligand concentration, cytoneme length, and contact sites (to match experimentally observed properties), based on our in vivo measurements after alteration of Vangl2 function (Fig. 3). We found that increasing ligand concentration (in these simulations by a factor of 10) within the morphogenetic field leads to an anterior shift of MHB in comparison to the control situation (Fig. 7G, H). We found that increasing lengths of cytonemes per cell after expression of Vangl2 in the source cells (by the experimentally determined factor of 33%) leads to a slight anterior shift of the MHB (Fig. 7I). Remarkably, lengthening of cytonemes and increasing the number of contact sites and the ligand concentration—comparable to the co-expression of Vangl2 and Wnt8a (Fig. 3D)—led to a significant broadening of the hindbrain territory (Fig. 7J). Notably, we also found the MHB becomes less distinct, (Fig. 7B), whereby there are considerably more cells exhibiting a fate incongruous with that defined by their position relative to the computed boundary. Finally, we asked the question if the observed phenotype observed after increasing the length of Wnt8a positive cytonemes (Fig. 7J) can be rescued by reducing the capability of cells to form cytonemes. Therefore, we addressed the question of how reduction of the mean cytoneme number would affect the positioning of the MHB in a morphogenetic field with increased Wnt ligand concentration and Vangl2 controlled cytonemes. We find that the MHB shifts back towards a posterior position, however, a full rescue of the phenotype could not be observed in our simulations (Fig. 7K). On the basis of the simulations, we predicted a strong increase in the range of Wnt signal activation as well as a loss of robust boundaries in the neighboring tissue, when Vangl2 function is accelerated in the Wnt8a source cells.

**Vangl2 activity influences neural plate patterning in zebrafish neurogenesis.** To test our prediction from the simulation, we analysed the consequences of Vangl2 function on the Wnt signaling range in zebrafish embryogenesis. In detail, we wanted to understand if the changes to cytoneme length and contact sites, as a result of Vangl2 alterations, impacted on neural plate patterning as suggested by agent-based simulation. First, we analysed the formation of the Wnt source tissue. The zebrafish embryonic margin functions as a major signaling source for Wnt and Fgf in the early gastrula. The transcription factor Notail (Ntl; ortholog in mouse TBXT, formerly known as T or brachyury) is essential for the induction of signaling factors such as Wnt8a and Fgf8a at the embryonic margin[56]. Ntl expression is detected early in development in Wnt8a positive mesodermal progenitor cells and is required for body axis formation. Wnt8a and Ntl act in a positive autoregulatory loop to reinforce their expression[57]. Therefore, we analysed the expression pattern of *ntl* at the embryonic margin after altering of expression of Wnt8a and

Vangl2. At 60% epiboly, the overexpression of *vangl2*; and more drastically, *wnt8a* with *vangl2*, caused an abnormal broadening and ectopic expression pattern of *ntl* (Fig. 8A). To link the phenotype observed after co-expression of *vangl2/wnt8a* to cytoneme appearance, we performed an experiment in which we reduce cytoneme formation by IRSp53[4K] overexpression in embryos expressing *vangl2* and *wnt8a*. We found that reduction of cytoneme number in Vangl2/Wn8a expressing embryos rescued partially the increase of the *ntl* expression domain (Fig. 8A).

Next, we mapped the expression pattern of neural plate markers. *gbx1* is a further direct Wnt signaling target in the gastrulating embryo and a marker for hindbrain identity[58]. *gbx1* mRNA expression can be detected at the marginal region at 7hpf (60–80% epiboly) (Fig. 8A). By measuring the intensity of the *gbx1* expression from vegetal to animal pole, we were able to measure the width of the expression pattern, as well as the sharpness of the MHB (Fig. 8B). Wnt8a expression leads to a broadening of the expression pattern of *gbx1* compared to control and a reduction in the sharpness of the MHB (Fig. 8A, B). Vangl2 alone has little effect on *gbx1* expression, with similar expression distance and sharpness of boundaries to control. However, *wnt8a* and *vangl2* overexpression together leads to a more exaggerated broadening to the *gbx1* expression domain pattern compared to control and *wnt8a* mRNA alone. This phenotype requires phosphorylation of Vangl2 as the Vangl2[10A] mutant shows a similar phenotype like in the control embryos. Our data suggests that the Wnt/β-catenin signaling range is broadened, most likely due to the increase of cytoneme length and number of contacts of cytonemes by Vangl2 as observed earlier (Fig. 3). In addition, there is a reduced sharpness of the midbrain-hindbrain boundary (MHB), as a result of more ectopic *gbx1* expression further from the original expression domain. As predicted from our simulation, in synergy, Wnt8a and Vangl2 leads to a broadening and less sharp *gbx1* expression domain, suggesting that the increase in cytoneme length and contact points may affect the signaling range and capability to signal to further cells. This has an impact on the specification of the brain primordia, for example, the *gbx1* positive hindbrain primordium. Our simulation predicts that a reduction of cytonemes leads to a partial rescue (Fig. 7K). Indeed, blockage of cytoneme formation by IRSp53[4K] overexpression rescues partially the phenotype observed in *wnt8a* and *vangl2* expressing embryos as previously observed (Fig. 8A, B).

By 24hpf, primordial brain boundaries have formed due to activation of specific markers for brain regions. *pax6a* is expressed at the forebrain primordium and at the hindbrain primordium and rhombomeres. Addition of Wnt8a posteriorizes the brain, leading to reduction of forebrain and midbrain primordia structures (Fig. 8C, D). The observed posteriorizing is an indication of increased Wnt/β-catenin signaling. *Vangl2* overexpression leads to similar anteroposterior patterning as control. However, overexpression of *wnt8a* plus *vangl2* leads to significantly reduced forebrain and midbrain structures, with complete abolishment of forebrain primordia structures in some cases. This is more severe than *wnt8a* alone. Emergence of

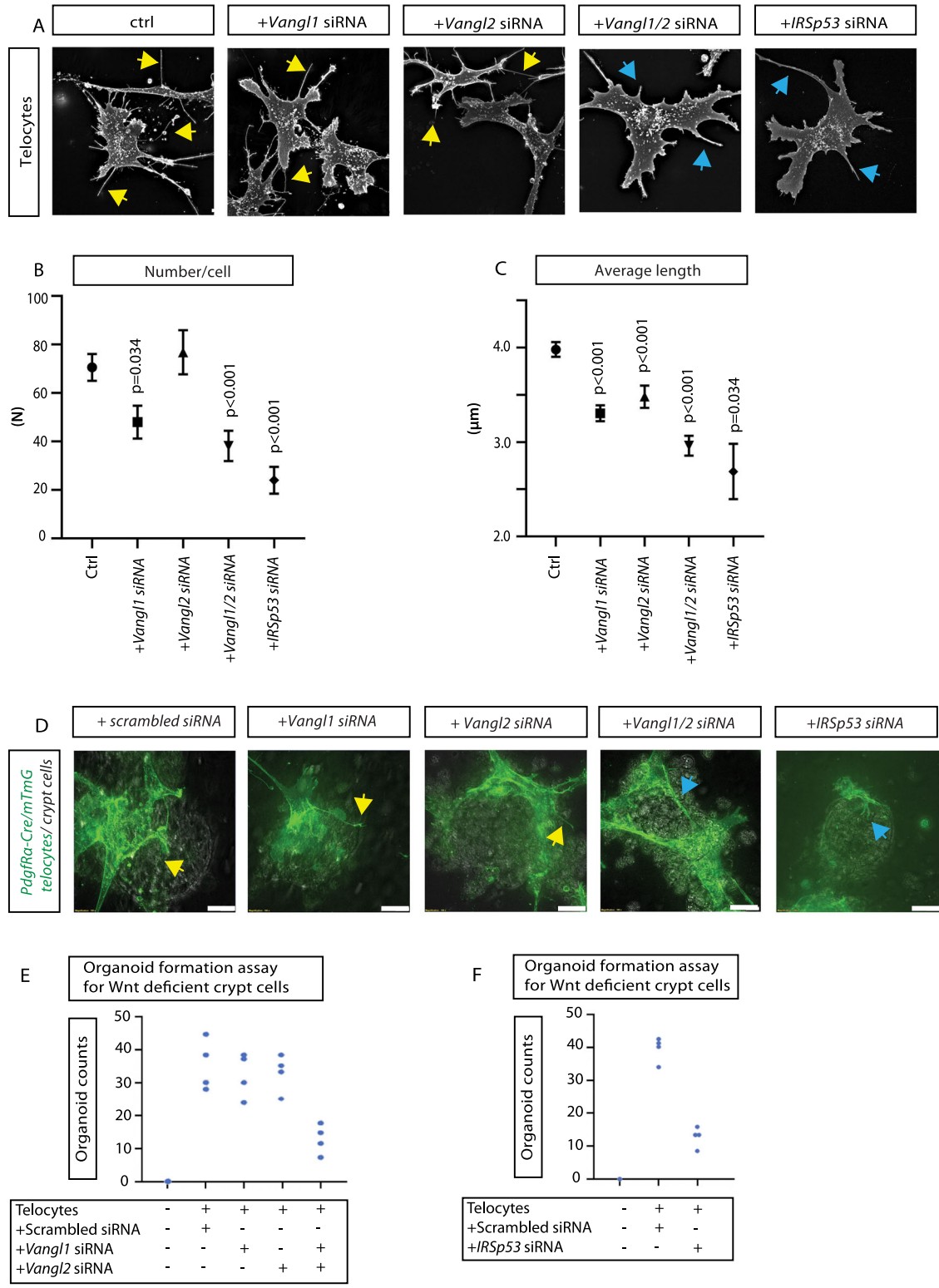

convergent extension (CE) phenotypes with addition of PCP *vangl2* is only evident after the addition of *wnt8a* (Fig. 8C, E). Therefore, as well as a CE phenotype, Wnt8a and Vangl2 causes a neural plate patterning phenotype, with reduction of forebrain and midbrain primordia structures. Finally, we asked if the observed alteration of AP patterning can be linked to cytoneme emergence. Therefore, we reduced the formation of cytonemes in the Wnt8a/Vangl2 expressing embryos by co-expression of

IRSp53[4K]. We find that the observed posterization phenotype is rescued (Fig. 8C, D). However, the CE phenotype shown by an open neural tube phenotype could not be rescued. We speculate that due to the reduction of filopodia formation per se, cell migration could be affected. Overall, Vangl2 alterations leads to changes in early neural plate patterning, as outlined by changes to *ntl* and *gbx1* expression. At later stages, CE and AP patterning phenotypes of the neural tube are present. We suggest that the

**Fig. 6 Vangl function in murine telocytes is required for the formation of Wnt deficient intestinal crypt. A** Telocytes stained with FITC-phalloidin. Yellow arrows indicate filopodia protrusions. Blue arrows indicate thicker protrusions. Scale bar = 10 μm. **B**, **C** Number of filopodia per cell and average length of filopodia in telocytes treated with control scrambled siRNA and siRNA for Vangl1, Vangl2 and Vangl1/Vangl2, IRSp53. **B** (n = 35, 21, 22, 27, 6 cells). **C** (n = 61, 50, 30, 30, 6 cells, n = 3488, 1796, 1964, 1079, 92 filopodia). Graphs represent mean and SEM. Statistical significance: p ≤ 0.05. **B**, **C** Two-sided Kruskal-Wallis tests with Bonferroni correction for multiple tests. SEM = 1. **D** Organoid formation assay: PDGFRα-cre/Rosa2-mTmG telocytes (green), co-cultured with Porcn$^{-/-}$- intestinal epithelial crypt cells (brightfield) in scrambled siRNA, Vangl1, Vangl2, Vangl1/Vangl2, and IRSP53 siRNA treated conditions. Micrographs were taken day 1 after co-culture, indicating that cells targeted with control or specific siRNAs were establishing extensive contacts with organoid epithelial cells. Arrows indicate long filopodia-like membrane protrusions in siRNA treated control group while in groups treated with Vangl1/2 or IRSp53 siRNA contacts are established via lamellipodia and thicker, telome-like structures. Scale bar = 10 μm. **E** Organoid formation assay of Porcn deficient crypt cells co-cultured with murine telocytes treated with control scrambled, Vangl1, Vangl2, or Vangl1/Vangl2 siRNA. Organoid counts were recorded per condition (n = 4 assays per condition). **F** Organoid formation assay of Porcn deficient crypt cells co-cultured with murine telocytes treated with control scrambled, IRSP53 siRNA. (n = 4 assays per condition). Source data are provided as a Source Data file.

observed effect on AP patterning of the zebrafish brain anlage is a result of increased Wnt cytoneme length and cytoneme contacts, impacting on an increased Wnt/β-catenin signaling activation range.

## Discussion

The PCP signaling pathway was initially characterized in *Drosophila* and orchestrates cell polarity across an epithelium[59]. For example, in Drosophila, PCP is responsible for the coordinated and consistent orientation of hairs and bristles. In vertebrates, PCP homologs regulate the orientation of inner ear sensory hair cells, hair follicles of the skin, and epithelial cells bearing multiple motile cilia. PCP is further involved in regulating cell migration as seen during convergent extension during gastrulation, differential adhesion across cells, orientation of cytoskeletal elements, and positioning of cell extensions, such as filopodia[7]. Core members of the Wnt/PCP signaling in vertebrates are, for example, Vangl and Ror2[60]. The transmembrane factor Ror2 serves as a Wnt co-receptor helping to relay the signal to Vangl. After activation by the Wnt5a ligand, Ror2 forms a ligand-receptor complex to which Dvl and CK1δ/ε is recruited. CK1δ/ε phosphorylates Dvl[61,62] and the two Ser/Thr phosphorylation clusters in the cytoplasmic N-terminal region of the Vangl2 protein[20,21]). In mouse, mutants for the core PCP component Vangl2 exhibit open neural tubes (craniorachischisis)[19,63]. Similarly, human mutations in both VANGL1/2 are associated with spina bifida[64,65]. In zebrafish, the *vangl2* mutant *trilobite* exhibits a broadened body axis, owing to similar defects in convergent extension (CE) movements during development[66]. Therefore, Vangl2 together with Ror2 take center stage in the PCP signaling pathway in vertebrates.

**Vangl2 and cytoneme formation**. Here, we demonstrate that Vangl2 has an essential function during formation of Wnt cytonemes and thus in the distribution of Wnt ligands across vertebrate tissues. The mechanism of cytoneme-based ligand transport has been observed in many tissues[11,15,67]. Initially, cytonemes have been described in various tissues of Drosophila transporting Fgf, Dpp, and HH signaling components[68–71]. Recent data suggest that cytonemes are also used to mobilize signaling components in vertebrates. Shh is transported by cytonemes in the chick limb bud and uses a Cdc42-independent mechanism[72]. Cytonemes are also fundamental for transporting Wnt signals. We have shown that cytonemal transport of Wnt8a is essential during neural plate patterning during zebrafish gastrulation[14]. Wnt is loaded on cytonemes and can be found at the cytonemal tip (Fig. 1). In chick, there is evidence that also the Wnt receptor Frizzled7 (Fzd7) is required for somite formation and Fzd7 puncta could be detected on cytonemes emitting from the ectodermal layer[73]. Activation of Wnt signaling in stem cells can be

similarly mediated via Wnt receptor positive cytonemes[74]. These findings are similar to observations in Drosophila, in which cytonemes containing Fzd receptors extend from myofibroblasts to pick up Wg signal from the wing disc[75]. However, the molecular mechanism underlying cytoneme formation is still unclear.

Here, we show that Vangl2 can be seen together with Wnt8a and Ror2 on cytoneme tips of PAC2 fibroblasts as well as zebrafish epiblast cells in vivo. These data are supported by observations in zebrafish epiblast cells and zebrafish hindbrain motor neurons, in which Vangl2 localizes to filopodia tips[25,26]. Similarly, in the mouse neural tube, Vangl2 and Frizzled3 are enriched on the tips of growth cone filopodia[23]. However, the function of Vangl2 in cytonemes is yet to be elucidated.

We provide evidence that Vangl2 regulates the appearance of Wnt-bearing cytonemes: Vangl2 activation induces specifically long, branched cytonemes, which carry Wnt protein at their tips. To reduce with Vangl2 function, we expressed the N-terminal truncated form of Vangl2 and the Vangl2$^{10A}$ mutant. Both tools have been shown to inactivate Vangl2 signaling[20,21]. We decided against the usage of the zebrafish *vangl2* mutant *trilobite* or a Morpholino-based knock-down approach, because our experimental strategy demanded the analysis of the cytoneme-generating cells without interfering with the cytoneme-receiving cells. Therefore, a mosaic expression strategy was preferable. To complement our analysis in zebrafish embryos, we knocked-down Vangl1/2 in mouse telocytes by an siRNA approach (Fig. 6). We find that reduction of Vangl signaling causes significantly fewer cytonemes, demonstrating a function for Vangl2 in cytoneme mediated Wnt transport. Besides its function in cell polarity and coordinated cell migration, Vangl has been suggested to play an important role in cellular protrusion formation. In *Drosophila*, knock-down of vangl and prickle lead to the formation of very few and short Fgf cytonemes[76]. In Vangl2$^{-/-}$/*Loop-tail* mouse mutants, filopodia were unable to extend[77]. In hippocampal neurons, Vangl2 was found to regulate dendritic branching, with Vangl2 knockdown leading to reduced spine density and dendritic branching[78]. In zebrafish, Vangl2 modulates the formation and polarization of actin-rich, large protrusions in ectodermal cells[25]. Vangl2 is enriched at membrane domains that are developing these large protrusions compared with non-protrusive domains. Interestingly, Vangl2 has been suggested to de-stabilize protrusions, whereas Fzd3a is required to stabilize the same extensions[26]. However, the nature of these protrusions is unclear. Here, we investigated the function of Vangl2 on cytonemes—small, slender protrusions, which form and retract within minutes and are loaded with signaling proteins. We show that Vangl2 function is essential for the formation of long Wnt cytonemes.

**PCP/Vangl2 signaling regulates JNK signaling**. The mechanism of how Vangl2 regulates cytoneme appearance is unknown. Our data suggests that cytoneme emergence can be modulated by an

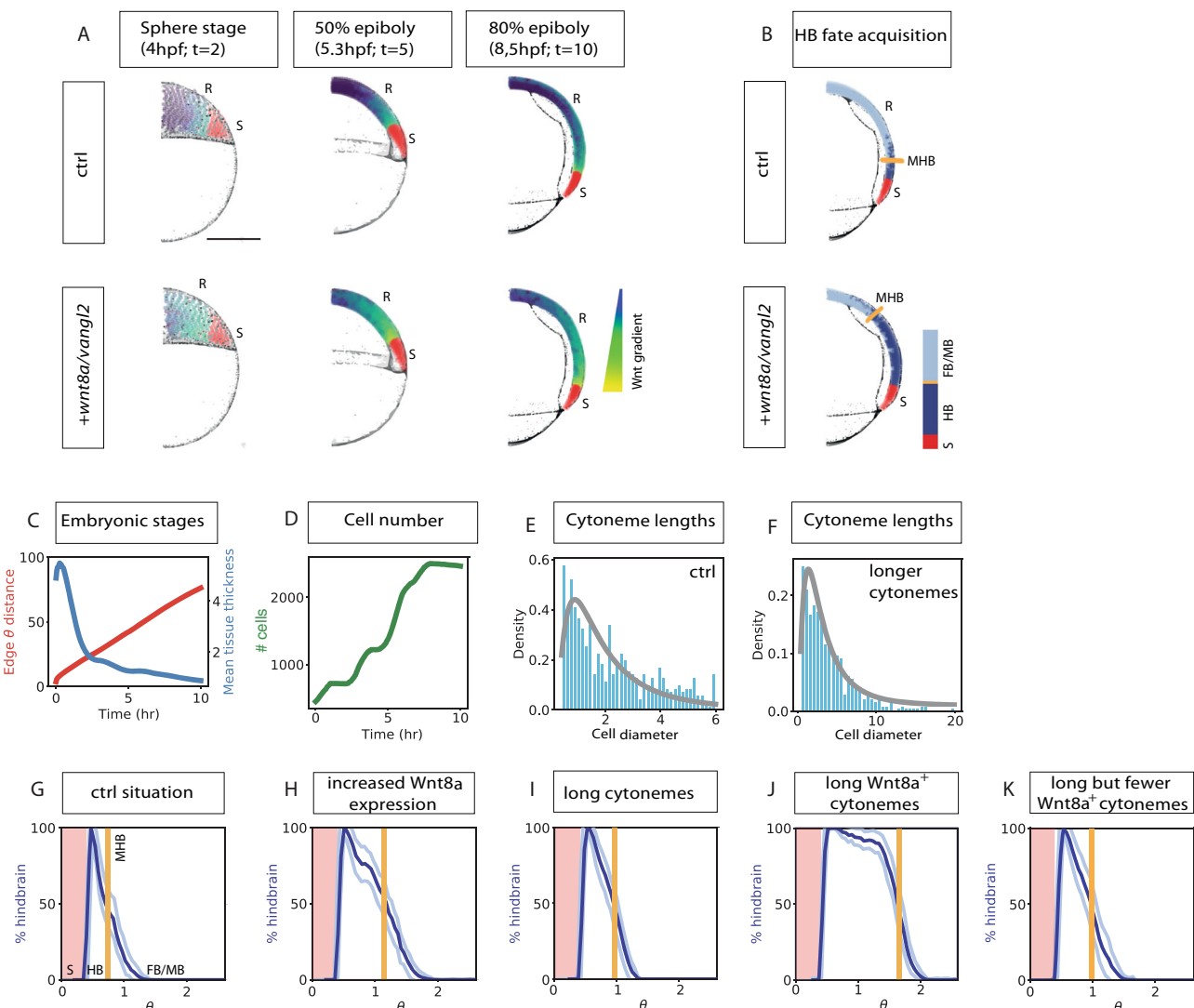

**Fig. 7 Agent-based modeling predicts an essential role for Vangl2 in Wnt-mediated tissue patterning during zebrafish gastrulation. A, B** Comparison of Wnt protein distribution in control zebrafish embryos with zebrafish embryos with Vangl2/Wnt8a overexpression. State of one realization of the model at the indicated time point (*t*): (Sphere stage = 4 hpf (hours post fertilization); 50% epiboly = 5.3 hpf; 80% epiboly = 8.5 hpf) for base parameter values and Wnt8a and Vangl2 over-expression parameters. Scale bar = 250 μm. For the 80% epiboly panel, the fate of the cells is also plotted in **B**. Source cells (S), are indicated in red. R = Receiving cells. The colors of the cells in anterior tissue correspond to the relative level of Wnt8 protein received (**A**). For ease of viewing, Wnt8a protein values have been normalized by the maximum value attained by any cell across the simulation and log transformed. In the cell fate diagram (**B**), cells acquiring a hindbrain (HB) fate are marked in dark blue, whilst those not acquiring a hindbrain fate are marked in gray. The orange line marks the estimated midbrain-hindbrain boundary (MHB). **C** Tissue growth properties—a single simulation. Note that the mechanics of the tissue growth are preserved across all conditions and so this graph is representative of all simulations. The red curve shows the proportion of the yolk that is covered by the tissue at the indicated times. The blue curve shows the mean tissue thickness (in terms of the number of cells). **D** Evolution of the cell number. E, **F** Histograms of cytoneme lengths at the final state of the simulation (normalized by cell diameter) for the control parameters (**E**) and with longer cytonemes (**F**). The gray curve shows a fit of the data to a log-normal distribution with means 2.0 (**E**) and 3.8 (**F**). **G–K** Distributions of cell fates over the angular polar coordinate at the end state of the simulation according to a hindbrain Wnt8a threshold of 100 (AU) for control parameters. The length of an arc of a circle is equal to ∅ is equal to 250 μm (**G**), Wnt8a overexpression parameters (**H**), longer cytonemes (**I**), longer Wnt8a positive cytonemes (**J**), long but fewer Wnt8a positive cytonemes (**K**). The dark and light blue curves respectively show the mean and standard deviations of the proportion of cell fates acquiring a hindbrain fate in 100 equi-spaced bins around the yolk over 100 model simulations for each condition. The orange line shows the estimated position of the MHB. FB/MB forebrain/midbrain. The red shaded area marks the position of the margin of wnt8a producing cells. Source data are provided as a Source Data file.

intrinsic signaling cascade and Rac/Jun N-terminal kinase (JNK) and Rho-associated kinase (ROK) participate in the Wnt-mediated PCP pathway. The Rac/JNK signaling cascade is an intracellular relay pathway and is essential in regulating both the cytoskeleton and cell adhesiveness[79]. At the core of this cascade are the stress-activated MAP kinase kinases MKK4 and

MKK7 that activate JNK to modulate cytoskeletal and nuclear events[80,81]. During dorsal closure in Drosophila, JNK signaling regulates the formation of actin and myosin dependent protrusions[82,83]. We have previously identified several members of the Rac/JNK signaling family as regulators of Wnt8a positive cytonemes[17]. We showed that overexpression of MKK4 and JNK3

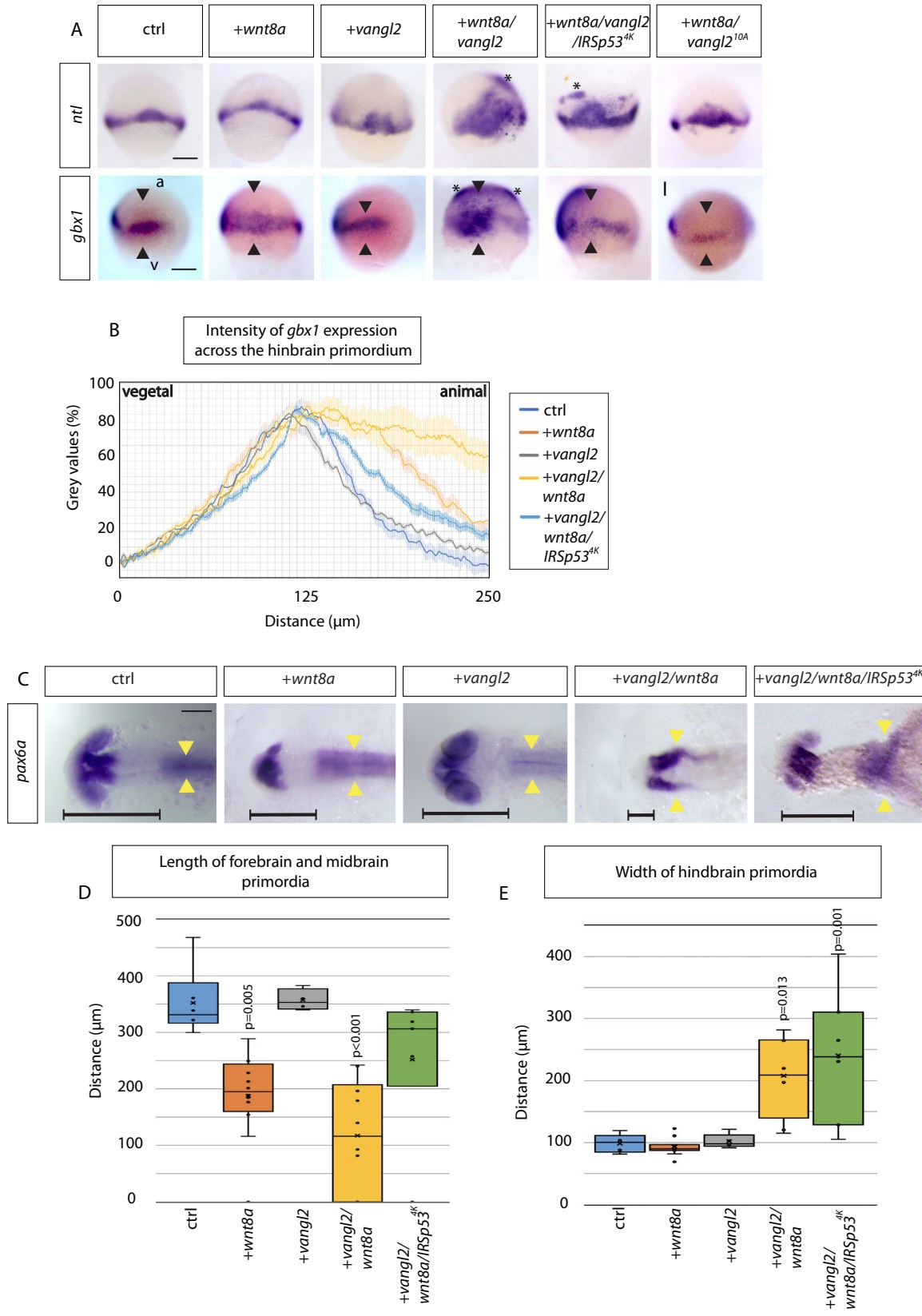

led to the formation of longer Wnt8a cytonemes in PAC2 cells. Furthermore, we have shown that blockage of Rac1 (and Cdc42) mediated signaling by ML141 leads to a collapse of Wnt cytonemes within 2h[14]. Here, we show that Vangl2-positive cytonemes are retracted within minutes after JNK signal inhibition (Fig. 4).

Indeed, there is evidence of Vangl2 mediated regulation of JNK activity. Vangl2/PCP signaling leads to activation of JNK and c-Jun by phosphorylation[80,84]. Furthermore, Vangl2 regulates cell adhesion by regulating Rac1/JNK activity at adherens junctions[85]. Similarly, it has been suggested that Vangl2 promotes phosphorylation of c-Jun and AP-1 in zebrafish[86]. It has been further

**Fig. 8 Vangl2 function is crucial for anteroposterior patterning of the zebrafish neural plate. A** In situ hybridization analysis of indicated markers in 60% epiboly zebrafish embryos (6.5 hpf) injected with *Wnt8a, Vangl2 and Wnt8a/Vangl2, wnt8a/vangl2/IRSp53⁴ᴷ, wnt8a/vangl2¹⁰ᴬ*. Scale bar indicates 200µm. (*ntl*: n = 35, 16, 6, 12, 13, 19 embryos) (*gbx1*: n = 6, 19, 3, 11, 22, 19 embryos). a = animal pole, v = vegetal pole. Black arrowheads indicate width of *gbx1* expression. Asterisks indicate ectopic expression. **B** Intensity of *gbx1* expression from in situ hybridization (gray values in %) across the hindbrain primordium from vegetal (v) to animal (a) pole in embryos injected with indicated constructs at 60% epiboly (6.5 hpf): Control- light blue, *wnt8*- orange, *vangl2* - gray, *wnt8a/vangl2*- yellow, *wnt8a/vangl2/IRSp53⁴ᴷ* - dark blue. **C** In situ hybridization to mark *pax6a* expression in the primordia of the forebrain and the hindbrain in embryos injected with mRNAs for the indicated constructs at 24 hpf: control, *wnt8a, vangl2, wnt8a/vangl2,* and *wnt8a/vangl2/ IRSp53⁴ᴷ*. Scale bar indicates 100 µm. Horizontal black line indicates length of forebrain and midbrain primordium. Yellow arrowheads show the width of the hindbrain primordium, indicating the extent of convergent and extension of these cells. (n = 13, 19, 8, 10, 7 embryos). **D** Box and whisker plot of length of forebrain and midbrain primordia in control (blue), Wnt8a (orange), Vangl2 (gray), Wnt8a/Vangl2 (yellow) and Wnt8a/Vangl2/IRSp53⁴ᴷ (green) 24 hpf larvae (µm). Measured from anterior forebrain *pax6a* expression to the position of the midbrain-hindbrain boundary (MHB) shown by horizontal black line (**C**). (n = 6, 12, 4, 10, 7 embryos). **E** Box and whisker plot of maximum width of hindbrain primordia in control (blue), Wnt8a (orange), Vangl2 (gray), Wnt8a/Vangl2 (yellow) and Wnt8a/Vangl2/IRSp53⁴ᴷ (green) 24hpf larvae (µm). Measured from maximum width of hindbrain *pax6a* expression shown by yellow arrowheads (**C**). (n = 5, 11, 5, 8, 7 embryos). **D**, **E** Data represented as box and whisker plots. Whiskers define the minimum and maximum values. Bounds of box indicate the 25th and 75th percentile. center line indicates the median. Cross indicates mean. Outliers and inner points shown. Statistical significance: p ≤ 0.05. **D**, **E** One-way ANOVA tests plus Tukey's post-hoc test. SEM = 1. Source data are provided as a Source Data file.

suggested that p62 is required to recruit and activate JNK through an evolutionarily conserved VANGL2–p62–JNK signaling cascade in Xenopus and human breast cancer cells[87]. We conclude that Vangl2/PCP activates Rac1/JNK signaling to regulate cytoneme generation in the Wnt source cell.

In addition to JNK signaling, Wnt/Vangl stimulation can also induce activation of RhoA-dependent signaling. RhoA/Rock regulate cell migration[88]—specifically convergence and extension (CE) movements—by mediating Wnt/PCP signaling in zebrafish and Xenopus gastrula[35,89–91]. Although there are reports of cross-regulation and synergism, RhoA/Rock signaling has been rather implicated in changes in cell shape, orientation, and polarity whereas Rac/JNK signaling is involved in filopodia formation[92]. This notion is supported by our observation, that JNK signaling is essential for controlling the formation of highly dynamic cytonemes, whereas RhoA/ROCK signaling seems to be more important for adjusting the number of filopodia[93] or cytonemes[14] presumably as an adaption to an altered cell morphology. Indeed, we found that Cdc42 can regulate the number of cytonemes[14] similar to RhoA, which regulates cell shape and protrusion lifespan in CE movements[92]. Therefore, we hypothesize that Vangl2 activates Rac/JNK signaling to regulate the elongation and collapse of Wnt cytonemes within minutes and, therefore, influences the dynamic distribution of Wnt ligands on cytonemes.

In addition to the intrinsic function of Vangl2 on Rac/JNK, it has been suggested that Vangl/PCP signaling influences protrusion formation by influencing the composition of the extracellular matrix in Drosophila and zebrafish[25,76]. In the zebrafish *vangl2* mutant *trilobite*, longer and thicker protrusions were observed. However, it is unclear if these extensions are filopodia, which are used for signaling or nanotubes forming intracellular bridges[94]. In Drosophila, it has been suggested that the PCP components prickle and vangl are essential for Fgf cytonemes formation. After blockage of prickle/vangl function, cytonemes were reduced in number and length when *pkRNAi* or *VangRNAi* were expressed[76]. In these flies, the composition of the extracellular matrix (ECM) was altered, suggesting that prickle and vangl are involved in maintaining normal levels of glypicans such as Dally, Dlp, and laminin. However, a molecular mechanism explaining the relation between PCP signaling and ECM composition is unclear.

**Vangl2 cytonemes and paracrine signaling and tissue patterning.** Recently, we compared modes of signaling transport, specifically we compared diffusion-based transport to cytoneme-regulated transport[95]. We found that cytoneme-based transport is required for patterning in quickly developing tissues such as during

zebrafish gastrulation. Diffusion-based distribution requires significantly more time to establish a robust pattern in the receiving tissue. In particular, the time taken for the distribution to form is inconsistent with the development time checkpoints during gastrulation. Thus, we here exclude diffusion as a transport mechanism for Wnt8a signaling in this context.

We further could show that Wnt distribution by cytonemes is essential during a very narrow time window of 2 h to achieve neural plate patterning during early zebrafish gastrulation[95]. This raised the question of how to reach a sufficiently high flux of Wnt protein into the field by fast cytonemes, given the fact that only a few cytonemes form per Wnt producing cell. Here we show that Vangl2 activation is a crucial regulator of cytoneme behavior regarding length, and growth properties, and the ability to branch out with the potential to contact multiple target cells. This observation became relevant for the generation of the morphogenetic field in our simulation. When the simulation assumed that all cytonemes deliver the ligand to a single neighboring cell and we take into account the actual number of cytonemes, we observed only minimal differences in the concentration of the ligand within the morphogenetic field. However, as soon as we allow multiple contact sites per cytoneme over a longer time, we increase the level of activation in the target cells combined with a wider patterning activation (Fig. 7). We concluded that for generating a signaling gradient employing the mechanism we describe it is essential to regulate all parameters of cytonemes. In particular, the net change in the flux of Wnt8a from producing to receiving tissue controls the range of the gradient and is directly proportional to the filopodia number and length but more importantly, also to the number of contact events per cytoneme. We observed that activation of Wnt cytoneme transport by Vangl2 and Ror2 led to significant upregulation of Wnt/β-catenin signaling in the neighboring cells, whereas blockage of Ror2 or Vangl2 function led to a strong reduction of paracrine Wnt signaling. Consequently, activation of Vangl2 and Wnt8a led to a synergistic upregulation of paracrine Wnt/β-catenin signaling and in posteriorization of the neural tube during zebrafish gastrulation. In future work, the predictions of our model could be validated by investigating paracrine signal activation and tissue patterning. We argue Vangl2 function is essential to control cytoneme length and contact events given the short time window in which Wnt signaling is required for neural plate patterning.

Cytonemes are taking center stage in cell–cell communication in invertebrates[67], and there is an increasing number of examples similarly describing the essential function of these signaling filopodia in vertebrates[15]. However, our understanding of how

these cytonemes are regulated is still in its infancy. Here, we describe Vangl2/PCP-induced cytonemes as transport carriers for Wnt8a in zebrafish. In cell culture experiments, we use PAC2 fibroblasts and HEK293T cells to provide further evidence for the importance of Vangl2-dependent regulation of cytonemes in Wnt trafficking. In addition, we show that human gastric cancer cells AGS process paracrine Wnt signaling via cytonemes, which are influenced by Vangl2 activity. Then, we use murine intestinal stroma cells, which express multiple Wnts to maintain the Wnt gradient operating in the intestinal crypt and provide evidence that also these Wnts utilize Vangl1/2-dependent cytonemes for their transport. Finally, we show that Vangl2 activity controls neural plate patterning in the developing zebrafish embryo by controlling Wnt8a cytonemes. In summary, we show that autocrine PCP pathway activation via Vangl2 induces Wnt cytonemes in the Wnt source cell to transmit Wnt to the neighboring cell to activate paracrine Wnt/β-catenin signaling. We propose that PCP signaling is a crucial mechanism for regulating the emergence of cytonemes to mobilize Wnt ligands in vertebrate development and tissue homeostasis.

## Methods

**Zebrafish maintenance and husbandry**. WIK wild-type and Tg(vangl2:GFP-Vangl2)[32] zebrafish (Danio rerio) were maintained as previously described at 28 °C and on a 14 h light/10 h dark cycle[14]. Zebrafish care and all experimental procedures were carried out in accordance with the European Communities Council Directive (2010/63/EU) and Animals Scientific Procedures Act (ASPA) 1986. In detail, adult zebrafish for breeding were kept and handled according to the ASPA animal care regulations and all embryo experiments were performed before 120 h of post-fertilization. Zebrafish experimental procedures were carried out under personal and project licenses granted by the UK Home Office under ASPA, and ethically approved by the Animal Welfare and Ethical Review Body at the University of Exeter. All mouse procedures were approved by the Singapore Health Services (SingHealth) Institutional Animal Care and Use Committee.

**Cell culture**. PAC2 zebrafish fibroblasts derived from 24hpf zebrafish embryos were maintained at 28 °C in Leibovitz L-15 media (Gibco). PAC2 cells[14] were kindly provided by Nicholas Foulkes (KIT) and Reinhard Koester (TU Braunschweig). HEK293T (Human Embryonic Kidney, CRL-1573) cells from the American Tissue Culture Collection, ATTC, Wesel, Germany were maintained at 37 °C with 5% CO$_2$ in DMEM media (Gibco). Primary gastric adenocarcinoma (AGS) cells were maintained at 37 °C with 5% CO$_2$ in RPMI-1640 media (Gibco). All media was supplemented with 10% FBS (Gibco) and 1% Pen/Strep (Gibco) and with L-Glutamine.

**Organoid formation of intestinal crypt cells**. Myofibroblasts were prepared from C57BL/6-Tg(PDGFRα-cre)[1Clc/J]/RosamTmG mice. To obtain stromal cells, intestinal crypts were removed from minced intestine by extensive pipetting in ice-cold PBS containing 2 mM EDTA. The epithelia-depleted stromal fractions were washed in PBS twice and then transferred into Advanced DMEM/F12 medium supplemented with Penicillin–Streptomycin, Glutamax (all from Invitrogen) and 2 mg/ml Collagenase/Dispase (Roche 11097113001). Tissues were digested on shaker at 37 °C for 3.5 h. Digestion was stopped by medium supplemented with 10% FBS. Non-digested debris were removed using a 70 μm strainer (Fisher Scientific), and then the digestion mix was centrifuged at $400 \times g$ for 5 min and washed once in PBS. Cells were then cultured in RPMI1640 medium supplemented with 10% FBS, 1% Penicillin–Streptomycin and 1% Glutamax (all from Invitrogen) until fully confluent (around 10 days). Minced intestine was depleted of epithelial cells by extensive pipetting in ice-cold PBS containing 2 mM EDTA. The stroma-enriched fractions were then washed in PBS twice and transferred into Advanced DMEM/F12 medium supplemented with Penicillin–Streptomycin, Glutamax (all from Invitrogen) and 2 mg/ml Collagenase/Dispase (Roche 11097113001). Tissues were digested on shaker at 37 °C for 3.5 h. Digestion was stopped by medium supplemented with 10% FBS. Non-digested debris were removed using a 70 μm strainer (Fisher Scientific) and then the digestion mix was centrifuged at $400 \times g$ for 5 min and washed once in PBS. Cells were then cultured in RPMI1640 medium supplemented with 10% FBS, 1% Penicillin–Streptomycin, and 1% Glutamax (all from Invitrogen) until fully confluent (around 10 days)[44].

As confluence of cultured cells was reaching 80%, they were transfected with Vangl1 and Vangl2 siRNAs (Dharmacon Cat# J-057276-09-0002 and Cat# J-059396-09-0002 respectively, at 10 nM) using siRNAmax reagent (Invitrogen Cat#13778–030). Knock-down was confirmed using RT-PCR and the primers shown in Source Data file. On day 2 post-transfection, myofibroblasts were mixed with Porcn-deficient intestinal epithelial and centrifuged at $400 \times g$ for 5 min to ensure close stromal/organoid contact. Thereafter, cells were mixed in 50%

Matrigel and cultured using RSPO1-supplemented medium. This ensures close cell-to-cell contacts decreasing need for migration. This assay measures Wnt secretion by stromal cells as all the necessary epithelial cell culture components are provided in medium except Wnts and the only source of Wnts is the added stroma. To ensure that both control-treated and Vangl1/2 siRNA-treated cells were in contact with organoids, cell cultures were photographed 24 h following co-culture. Organoid counting was performed at the time point when the group containing no stromal cells had no surviving organoids left (the end of day 3/beginning of day 4 of co-culture). siRNA-transfected cells were imaged using the Olympus Live Imaging system IX83. Acquired 3D image stacks were de-convoluted using the software cellSens Dimension (Olympus) and are presented as maximum intensity projections.

**Transfection and plasmids**. Cells were transfected using Fugene HD Transfection Reagent (Promega) and imaged after 24–48 h. Formation of filopodia and cytonemes were unaltered in cells transfected with a control construct compared to non-transfected samples. For the STF reporter co-culture assay, cells were transfected and after 24 h, re-trypsinised to co-culture cells.

Plasmids were used for transfection and to make in situ probes and mRNA for microinjection into zebrafish embryos. The following plasmids were used:

GPI-anchored mCherry in pCS2+(Mem-mCh)[96]; zfGap43-GFP in pCS2, xRor2 in pCS2+, XRor2[3i] in pCS2+ and xRor2-mCherry in PCS2+[17]; zfWnt8a ORF1-mCherry in pCS2+[58]; zfWnt8a ORF1-GFP in pCS2+[14]; meGFP-Vangl2 in pcDNA3.1[26]; zfStbm (Vangl2), in pCS2+, (Addgene, #17067); Stbm-DeltaN-6myc in pCS2+[86]; IRSp53[4K]; Vangl2 S5~17A::S76~84A[21]; JNK KTR-mCherry[38,39]; 7xTRE Super TOPFlash-NLS-mCherry (STF-mCherry)[42]; Lrp6-GFP[97].

**Antibody staining**. The IF staining were performed according to the published protocol[98]. In particular, PAC2 and AGS cells were plated on coverslips and depending on the experiment transfected with a zfWnt8a-mCherry plasmid using Fugene. Twenty-four hours later, cells were washed once in prewarmed PBS, then fixed using 0.25% glutaraldehyde and 4% PFA (for anti-Wnt8a antibody) or 0.2% glutaraldehyde (for anti-Vangl2 antibody) at 4 °C. Cells were washed in PBS, then blocked and permeabilised using 0.1% Triton X-100, 5% rabbit or goat serum, and 0.2 M glycine for 1 h at room temperature. Cells were washed again, then incubated with anti-Wnt8a antibody (1:50 dilution, catalog number: MBS9216179, MyBio-Source) or anti-human Vangl2 antibody (1:50 dilution, catalog number: SAB2501092, Sigma) overnight at 4 °C. Coverslips were washed, incubated with an anti-rabbit secondary polyclonal antibody conjugated to Alexafluor 488 or a donkey anti-goat AlexaFluor 647 (both 1:1000 dilution, Abcam, catalog number: ab150077 and ab150135, respectively) and Phalloidin TRITC/FITC (Merck) for 1 h at room temperature. Coverslips were washed and mounted using ProLong Mountant (Invitrogen).

**Image acquisition**. Cells and zebrafish embryos were imaged on a TCS Leica SP8 confocal microscope, using 63× dip-in objective. Cells were seeded on plastic 35 mm dishes and embryos were mounted in 0.7% low-melting point agarose in plastic 35 mm dishes and filled with Danieau's solution.

**Fluorescent intensity along cytoneme**. FIJI software was used to plot the pixel intensity along a selected line within an image. A line was assigned from the base to the tip of cytoneme as defined by a membrane marker and the pixel intensity was plotted. The line selection was copied to the ROI manager and the measurement repeated for the second channel.

**Fluorescence cross-correlation spectroscopy (FCCS)**. PAC2 cells were plated on coverslips and transfected with GFP-Vangl2 and Ror2-mCh. After 24 h, cells were treated with 500 ng/ml Wnt8a or Wnt5a mouse recombinant protein (R&D systems). Coverslips were inverted on to a depression slide containing media with or without mouse Wnt recombinant protein. A drop of water was added to the top of the coverslips and a focused laser spot was scanned across the membrane for 16 s while the intensity was measured as a function of time by using the Leica Sp8 HyVolution FALCON single-molecule detection unit. The intensity time traces were then time correlated. A "diffusion with triplet" fit model of the autocorrelation function and cross-correlation function was obtained by usage of the LAS X software.

**Number and length of cytoneme—FIJI analysis**. Membrane protrusions marked with membrane marker were assigned as filopodia. Membrane protrusions marked with membrane marker and positive for tagged Wnt8a were assigned as Wnt8a-positive cytonemes. In tissue culture analysis, PAC2 fibroblasts were transfected with the indicated constructs to mark cytonemes. In zebrafish embryos, cell clones were generated to analyze cytoneme formation within the live, intact embryonic tissue. Therefore, indicated mRNAs were injected into in 1 out of 8–16 blastomeres. This generates isolated fluorescently labeled cell clones, so that filopodia and cytoneme protrusions can be visualized and analyzed. Cells at the edge of the clone and isolated cells, where cytonemes can be seen, are measured for analysis. Control embryos were injected with mem-mCh and Wnt8-GFP. If not stated

differently, approx. 50 cells in 5 embryo or 30 cells of three independent transfection experiment were randomly chosen for analysis. In tissue culture as well as in the zebrafish embryos, the numbers and lengths of filopodia and cytonemes were measured manually from base to tip of protrusions in FIJI. Continuous nanotube structures from cell to cell or thicker membrane extensions were discounted.

**Microinjection of mRNA and DNA constructs.** Capped sense mRNA was generated from linearized plasmid using the mMessage mMachine SP6 & T7 Transcription Kits (Invitrogen). mRNA was microinjected at the 16-cell stage at 200 ng/μl, to generate clonal expression, then imaged live intact embryos from 50% epiboly. GFP-Vangl2 DNA, and mRNA was microinjected at 2–4 cell for clonal expression for in situ experiments.

**KTR-mCherry based JNK reporter assay and analysis.** HEK293T cells were transfected with JNK reporter KTR-mCherry and either/or, GFP-Vangl2, xRor2 and Stbm-DeltaN-6myc. Wnt5a and Wnt8a mouse recombinant protein was reconstituted at 100 μg/ml in PBS and 0.1% BSA (Biotechne). 500 ng/ml Wnt8a and Wnt5a mouse recombinant protein was added and incubated at 37 °C for 24 h before imaging. For analysis, mean gray values were recorded for 3× ROI in the nucleus and 3× ROI in the cytoplasm per cell and an average was taken. The ratio of cytoplasmic/nuclear signal was then recorded and normalized to 1 for the control.

**Inhibitor treatment.** Cells were transfected as described above. Twenty micromolar of JNK inhibitor SP600125 (Sigma), dissolved in DMSO, was added to the media. Cells were imaged immediately and 1 and 2 h after treatment. Y27632 (ROCK inhibitor) was dissolved in water and added to the media at 10 μM for 1, 5, or 24 h.

**STF reporter co-culture assay and fluorescent intensity analysis.** AGS cells were transfected with either STF reporter 7xTRE Super TOPFlash-NLS-mCherry/ Lrp6-GFP or plasmids of interest. After 24–48 h, cells were then trypsinised and seeded together for another 24 h. STF reporter expression was imaged on a Leica widefield microscope. Nuclei STF reporter expression was thresholded in FIJI and relative fluorescence of nuclei was measured. A nuclear DAPI staining was used to ensure that equal number of cells were present in each frame analysed.

**Generation of in situ probes and in situ hybridization.** *notail*, *gbx1*, and *pax6a* digoxigenin and FITC antisense probes were generated from linearized plasmids using an RNA labeling and detection kit (Roche)[99]. Probes were purified on ProbeQuant G50 Micro Columns (GE Healthcare).

Embryos were microinjected at 2–4 cell with mRNA and Vangl2 DNA and let to develop. Embryos were fixed with 4% PFA at 60% epiboly or 24hpf. Embryos were dehydrated in 100% MeOH and dechorionated. In situ hybridization was carried out as previously described[14]. In situ embryos were imaged on a stereo microscope with uplighter in 70% glycerol. 24hpf embryos were deyolked and flat-mounted under coverslips in 70% glycerol. The total length of the forebrain primordium expression of *pax6a* and negative stain of midbrain primordium before the start of hindbrain *pax6a* expression was measured in FIJI. Maximum width of hindbrain primordia was also measured in FIJI. Embryos with evidence of necrosis in the brain were discounted. Embryos did not develop further than 24 h after injection.

**Intensity of *gbx1* expression analysis.** FIJI software was used to plot the pixel intensity along 3× lines spanning *gbx1* expression from the animal to vegetal pole at 60% epiboly. Expression level by pixel intensity and distance of expression domain was recorded and averaged per embryo. Data were normalized to maximum control gray value and expression distance was aligned for comparison.

**Statistical analysis.** Statistical analysis carried out using IBM SPSS Statistics 26. Normal multiple comparisons were tested with one-way ANOVA plus Tukey post-hoc test. Non-normal multiple comparisons were tested using Kruskal–Wallis tests (two-sided) including Bonferroni correction for multiple tests if appropriate. If not explicitly stated otherwise statistical significance: * ≤ 0.05, ** ≤ 0.01, *** ≤ 0.001.

**Image and movie production.** Images were visualized and analysed in Leica LAS X software (3.7.2.22383) and FIJI (ImageJ 2.00-rc-59/1.53c). Movies were produced in Imaris 9.0.0 (Bitplane). Movies were produced at three frames/second. Time stamps and scale bars are also present on the movies for reference. Imaris 9.0.0 was further used to produce 3D renders using "Blend" and "Normal Shading" mode. 3D movies were made using the "Animation" option.

**Computer simulations.** The expansion of the neural plate was modeled in silico via agent-based simulation in the Chaste[52,53] C++ package on a HP Z840 workstation over an Intel Xeon E-series architecture. Forces between cells were defined via a Delaunay–Voronoi triangulation[100,101]. Radially acting forces resolved along the outward pointing normal of a circle were used to represent the surface of a yolk over which the tissue grew. A similar, but inwardly directed force was used to promote intercalation and convergent expansion around the yolk. Cell division followed a renewal process with uniformly randomly drawn division times (reset following a mitotic event). Cells underwent apoptosis following a Poisson process with a pre-defined probability per unit time.

Prior to simulation, the number of cytonemes for each source cell was set following a Poisson distribution with the experimentally observed values. The tip of each cytoneme was loaded with a uniform value of Wnt. At each subsequent time step, cytonemes were either expanded or fully retracted. During growth, cytonemes grew by an amount drawn from a normal distribution with mean equal to the experimentally observed (but scaled) values. Negative sample values were discarded and redrawn. The angle of cytoneme growth was drawn from a normal distribution centered on the tangent to the yolk (so that cytonemes grew preferentially towards the receptive tissue). Cytoneme retraction followed a Poisson process with uniform probability across all cytonemes. Cytonemes exceeding a maximum length as well as cytonemes with no remaining tip Wnt were also retracted. Retracted cytonemes were replaced instantaneously with another cytoneme with zero length at the same source cell and their tip Wnt values were reset, so that the total number of cytonemes in the simulation was conserved. Cytoneme retraction and growth properties were set according to experimentally observed values where data were available or were selected, so that simulations with control values were matched to control experiments.

At each time step, any cytoneme tip in contact with a receptor cell, (defined as being when the Euclidean distance between the tip and cell centre fell below the cell radius), deposited a fraction of its tip-loaded Wnt to the receptor cell in a probabilistic fashion according to a Poisson process. Following successful deposition, the Wnt values of both the cytoneme tip and receptor cell were updated. The deposition fraction was set so that control cytonemes contacted only one cell, whereas Vangl2 over-expressed cytonemes could make up to five contact events (though these could potentially be with the same cell). Wnt levels in the receptor cells decayed exponentially with constant rate between contact events.

At the end of the simulation (at $t = 10$ h), receptor cells were marked with either a hindbrain or non-hindbrain fate according to whether or not their Wnt level was above a threshold of 100. Following this, cell fates were binned according to the cells' angular polar coordinate around the yolk, with the origin set to be the leading edge of the source margin. Within each bin, the proportion of cells acquiring a hindbrain fate in each bin was calculated. These data were then averaged over the same bins for 100 simulations per condition. The MHB was defined to be the first bin for which this averaged proportion fell below half (so that without a bin, the majority of cells did not acquire a hindbrain cell fate). All simulation results were analysed in Python 3.6.3 (installed with Anaconda, https://www.anaconda.com) using the standard NumPy and SciPy libraries.

**Reporting summary.** Further information on research design is available in the Nature Research Reporting Summary linked to this article.

## Data availability

The FCCS data that support the findings of this study are available in Dryad, Dataset https://doi.org/10.5061/dryad.cfxpnvx4p. Additional data that support the findings of this study are available from the corresponding author upon reasonable request. Source data are provided with this paper.

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

## Acknowledgements

Research in the S.S. lab, including L.B., is supported by the BBSRC (Research Grant, BB/S016295/1 and an Equipment grant, BB/R013764/1) and by the Living Systems Institute, University of Exeter. S.R. is supported by the MRC (MR/S007970/1). Studies in the D.M.V. lab are supported by the National Research Foundation of Singapore and National Medical Research Council under its STAR Award Program as well as TIER3 grant MOE2016-T3-1-002. B.D.E. was generously supported by the Wellcome Trust Institutional Strategic Support Award (Grant number 204909/Z/16/Z). K.C.A.W. is supported by MRC Fellowship MR/P01478X/1. We would additionally like to thank Alex Fletcher (University of Sheffield) for assistance with the implementation of the agent-based model and Ned Boulter for programming python wrappers to facilitate parallel processing and analysis of the model simulations. For technical help, we would like to thank Jordan Kent (STF-assay), and Alexandra Mader (ISH analysis). We would like to thank Cecilia Moens (Fred Hutchinson Cancer Research Center) and Steve Wilson, Masa Tada (UCL) for providing plasmids; Trevor Dale and Toby Phesse (ECSCRI, Cardiff University) for providing the gastric cancer cell lines. Furthermore, we would like to thank Chan Yarn Kit for contributions to Vangl1/2 siRNA work and the entire Scholpp lab for critical comments on the manuscript. We would like to thank the Aquatic Resources Centre (ARC), the LSI Tissue culture facility, and the Bioimaging Centre, Exeter for excellent technical support.

## Author contributions

L.B. and S.S. designed, performed and analyzed all experiments except where noted and wrote the manuscript. S.R. performed the IF staining and the STF assay. G.G. and D.M.V. performed the intestinal organoid studies, B.D.E. and K.C.A.W. designed and performed the simulations.

## Competing interests

The authors declare no competing interests.
