## [Peer Review File · Nature Communications]

REVIEWER COMMENTS

Reviewer #1 (Remarks to the Author):

This work from the Scholpp lab continues their impressive and insightful investigations into the way that Wnts are distributed by cytonemes. There is much to like and admire in this installment, especially in the way that it progresses from identification of a role for Vangl2 in cell culture, to a mathematical analysis that predicts how perturbations to cytonemes in response to changes in the Vangl2 pathway might affect development of zebrafish embryos, and most importantly to direct experimental tests of these predictions. I enthusiastically recommend it for publication. My suggestions for clarification/improvement are minor.

Figure 1 Legend, text, Methods do not describe many essential details such as number of samples, whether they are fixed or unfixed, how fluorescence intensity was scaled, whether cytoneme position is shown tip to cell or cell to tip, length variation, ...

Conditions for controls for transfections and cytoneme length measurements should be described better. Do cytonemes change at different times of culture, after exposure to transfection reagents?

If I understand correctly, the cytonemes that are tabulated in Fig 2 are filopodia that contain Wnt8a. For the experiments in Fig 2, is the effect only these filopodia or all filopodia? Is cell viability, mitotic rate, or any other measure of cell vitality affected?

Figure 3 Is the method for embryo analysis described? Are these measures of intact embryos or dissociated cells from embryos? How many fluorescent-positive cells did the embryos have? How were the cells chosen for analysis? Are the controls uninjected or sham-injected? Do I understand correctly that here the definition of cytonemes is not the subset of filopodia that contain Wnt? To what extent are the data for cell culture and embryos consistent or inconsistent?

Figure 4B,C Is the greater variation in cytoplasmic/nuclear distribution responsible for the apparent increase for the Ror2/Vangl2/Wnt experiments? What would violin plots look like?

Line 248 Please explain why the mouse intestinal crypt was chosen and is presumably a better system for the experiments on Wnt transport than the zebrafish embryo.

Figure 6 Does this system provide the authors with the possibility of observing if the cytonemes are polarized in a particular direction or toward a particular field?

Figure 7 Can the authors provide a distance scale so we can understand the dimensions of Wnt signaling being modeled? Also a developmental time scale so we have a context for the cytoneme protrusion/retraction rates?

Figure 8 Can the authors provide some validation for the general state of these embryos – cell death/cell division/cell shape? Do they develop further?

Reviewer #2 (Remarks to the Author):

The manuscript by Brunt et al describes a finding that Vangl2 could regulate the formation of Wnt8-positive cytonemes (signaling filopodia) and thus have a positive function in the Wnt ligand dissemination and paracrine Wnt/ β -catenin signaling. Although this story is interesting, the role of Vangl2 in regulating filopodia has been documented (Love et al, Development 2018) in a clean genetic model (Vangl2 trilobite mutant zebrafish) rather than by overexpression experiments mainly used in

this study. I am actually very surprised that trilobite mutant was not analyzed in this study. The role of Vangl2 in initiating filopodia has also been demonstrated in mammalian cells (Shafer et al, Dev Cell 2012). In addition, the data presented are largely association, often time no definitive link can be established. There are some fundamental issues to be addressed, the specific points are listed below.

1. Obviously, the authors misunderstood the relationship between Ror2 and Vangl2 or they didn't read the literature carefully. They claimed that the kinase domain of Ror2 phosphorylates Vangl2 (page 3, line 68), however, there was no such evidence. Instead, many papers have shown that Vangl2/Vang is phosphorylated by casein kinase 1 (Gao et al, Dev Cell 2011; Kelly et al, Cell Rep 2016; Yang et al, Cell Res 2017; Strutt et al, Elife 2019).

2. Page 3, line 72-73, they described that "During zebrafish gastrulation, Vangl2 is asymmetrically localised at the plasma membrane and localizes to growing filopodia²³". I could not find the picture in the cited reference regarding the filopodia localization of Vangl2, please point it out. In their own studies, the overexpressed Vangl2 could be observed in the cytoneme (Figure 1D). But according to the signal in the cell body, Vangl2 is everywhere rather than the normal localization on the cell surface. Also for the clustering of Vangl2 and Wnt8a in the cytoneme, how to exclude the artifact of overexpression? Is there any evidence showing the cytoneme localization of endogenous Vangl2, which is more physiologically relevant for Vangl2's function?

3. Figure 1H-L, how the fluorescent intensity among different samples are normalized? The number of filopodia (N=5) measured is too low to draw a firm conclusion. Figure 1D and 1J, when GFP-Vangl2 and Ror2-mCherry are co-expressed, a beads-on-string pattern could be observed. If that is the case, why fluctuation of signals along the cytoneme was not detected.

4. Page 5, line 128-129, they found "Activation of Vangl2 caused a 58.5% reduction of cytonemes per cell". "The number of cytonemes are unaltered" by Ror2 transfection. For filopodia, Ror2 expression significantly increased the number of filopodia but Vangl2 decreased it (Supplementary Figure 3G). Why do Ror2 and Vangl2 have opposite effects on filopodia? The Ror2/Vangl2 PCP signaling is specific for Wnt8-positive cytoneme? Are Ror2 and Vangl2 expressed in Wnt8-negative cytoneme?

5. Figure 2D and F, with two channels, how do they know the cell is positive for three overexpressed genes (Wnt8a, Vangl2 and Ror2)? Supplementary Figure 3A-F, the intensity is too low to see the length and number of cytonemes in these cells.

6. They further used N-terminal deletion mutant of Vangl2 (Δ N-Vangl2) lacking the two Ser/Thr phosphorylation clusters to block Vangl2-mediated signaling. When I looked into the reference they cited (Gao et al., 2011) and a further in vivo studies of Vangl2 phosphorylation (Yang et al. Cell Res 2017), I could not find any evidence to support their conclusion that Δ N-Vangl2 could block Vangl2-mediated signaling. These papers, particularly the later one, showed dominant negative effects of phospho-mutant Vangl2 which appeared to affect more than Vangl proteins. Why not knock out Vangl1 and Vangl2 in the PAC2 cells to have more clean and convincing data, or at least use phospho-mutant Vangl2.

7. Figure 3, studies in zebrafish embryos. As mentioned earlier, the Vangl2 trilobite mutant zebrafish has been widely used for PCP studies. This is the most reliable way to study Vangl2's function in cytoneme.

8. Figure 3A-C, the signal intensity is much higher in B compared to A and C. They need to be shown at similar level. I can argue that the longer cytonemes in B are simply due to the higher overall fluorescent intensity.

9. Figure 4A, B. To be consistent with previous experiments, PAC2 cells should be used for KTR reporter experiment? What's the effect of Wnt5a and Wnt8a alone?

10. Figure 4F, it seems the length of cytonemes decreased over the time in the Vangl2 group and it is similar to the pattern of Vangl2 + SP group. How to explain this?

11. Figure 5A-C, the Wnt signaling activity experiment. This is only an association between the upregulation of Wnt signaling and Vangl2-regulated cytonemes. I can argue that this is due to the increased secretion of Wnt proteins somehow triggered by the Wnt8a/Ror2/Vangl2 overexpression in source cells or even indirectly through other mechanisms, for example, Vangl2 or Ror2 might stabilize Wnt8a or promote maturation of the ligand, why necessary through cytonemes? More importantly, if their model is true, does Vangl2 mutant animals (drosophial, zebrafish, or mouse) show the downregulation of canonical Wnt signaling? The in vivo evidence will be very critical for the conclusion.

12. Figure 6B, C, "the number of filopodia was significantly reduced after Vangl1 knockdown and even more markedly reduced after double knockdown of Vangl1/Vangl2". But as shown earlier in supplementary Figure 3G, the number of filopodia by Vangl2 overexpression is also reduced. How to explain this discrepancy?

13. Figure 6E, F. Fewer organoids formation is not a direct readout of downregulated Wnt signaling. Few organoids could be due to many reasons, for example, PCP may be important for telocyte's normal function or morphology. Again, this is only an association.

14. Figure 7, the authors tested cytoneme model, how about the diffusion model? If Wnt ligand transport is independent of cytonemes and just relies on secretion and diffusion, is there any difference compared to cytoneme simulation? If does, how to distinguish them experimentally?

15. Figure 8, "Vangl2 activity in the Wnt source cells regulates neural plate patterning in zebrafish neurogenesis" is an overstatement. It is possible that the change of expression patterns of these markers is secondary to a CE defect. Since overexpression Vangl2 is well known to cause CE defect, how did they distinguish the direct effect of Vangl2 on PCP and indirect effect through cytoneme on canonical Wnt signaling? Please clearly specify the region of Wnt source cells and receiving cells, is the phenotype restricted to the receiving cells? This part of data is not convincing, difficult to correlate it with cytoneme.

How Wnt proteins can spread in the extracellular space is an important but also controversial topic. A very recent study excluded all the previously proposed mechanisms of Wnt transport (e.g., lipoprotein particles, exosomes, chaperone) and demonstrated a new mechanism via glypican shielding (McGough et al, Nature 2020). However, whether cytoneme is indeed essential for Wnt dissemination is unclear due to the lack of genetic tool and evidence. Given the tremendous efforts made by this group on the Wnt cytoneme in the past, convincing genetic evidence is expected.

Reviewer #3 (Remarks to the Author):

The m/s by Brunt et al focusses on the role of cytonemes and their regulation by Vangl/Wnt signalling. The authors show a role for Vangl2 in Wnt distribution in zebrafish fibroblasts in culture and in vivo in gastrula stage embryos. In addition, Vangl2 activation appears to result in longer and fewer projections in zebrafish and in cultured human gastric cancer cells and mouse intestinal telocytes. Reduced Vangl leads to shorter cytonemes and reduced canonical Wnt signalling. The authors then modelled the function of Vangl2 in zebrafish gastrulation and predicted a shift of the morphogenetic gradient and altered neural tissue patterning, and tested these on the zebrafish neural plate. Overall, the manuscript reports interesting findings. However, the current data do not fully support the conclusions. Some additional analysis is needed to conclusively demonstrate a role for cytonemes in

Vangl/Wnt signalling in cultured cells and in embryos.

Major comments:

- With regard to cytoneme contacts between cells: In some instances, it is hard to discern if the contacts are actually there between cells. Showing 3D rotations of cytoneme contacts between two cells would be more conclusive (e.g. for Figure 3 I).

- There are only 1-2 cytonemes per cell. That being the case, it is difficult to assess a change from 2 to 1 (e.g. in DN-Vangl expressing cells,; Figures 2 and 3). Because the cytoneme N's are not meaningful and length ranges from 5-20 um, an alternate way of assessing contacts is required.

The perdurance of individual contacts could be a better measure of stability. For instance, to demonstrate that cytoneme stability is regulated by Vngl2, in addition to length and numbers per cell, the authors should assess the DURATION of each contact. If the cytonemes are severed at a different point along the projection length (from the contact location) that can be indicated, and regardless of breakpoint location, the time of severing (before cytoneme retraction) can be taken as the end point of contact.

- Fig 4b v and vi show cells with both nuclear and cytoplasmic expression or cytoplasmic expression alone (of the JNK signalling reporter).

These should be discussed and the numbers expressing nuclear, cytoplasmic and both should be shown. In addition, some cells (Fig 4 iv and vii) appear multinucleate. Why is that so? A membrane marker would be useful here.

- Figure 6: The telocytes where Vngl1 or 1+2 are knocked down show a cell shape change from spindle to more lobed morphology. Perhaps this underlies the defect in organoid formation in co-cultures with intestinal epithelial cells, rather than arising from cytoneme contact defects. How do the authors distinguish cytoneme defects from general morphological defects in these experiments? Other manipulations that lead to cell shape change (without affecting contacts) could be used as controls.

- The modelling of neural plate patterning (Figure 7) and test of the model in embryos (Figure 8) is interesting. The data presented show a role for vangl and wnt8 in hindbrain gbx1 expression and neural plate patterning; however, these do not conclusively show that the effect of vangl/wnt8 is cytoneme dependent. A more direct/local perturbation of cytonemes and its effects on Wnt distribution is needed to demonstrate this link.

This is important because the movies of embryos don't show as marked a change in vivo as the cultured cells e.g., upon DN-Vangl injections (e.g., Movie 5 versus 8).

Minor comments:

- In many cells, Wnt8 is seen not only at the tips but also along the length of the projections (e.g. Fig 1). These should be indicated with smaller arrowheads, to distinguish from the ones at the cell tips.

- Figure 4A; typo - change activitiy to activity

- Check and correct the manuscript for syntax/grammar and typographical errors.

Reviewer #4 (Remarks to the Author):

Long distance communication mediated by filopodia-like protrusions is an emerging field. There is increasing evidence that it plays important roles in tissue patterning by presenting ligands or receptors to distant cells. Cytoneme is one of the cellular processes shown to carry Wnt8a and to activate the pathway and pattern the neural tissue. Cytoneme formation has been associated with Wnt8a-mediated Ror2 activation. The current study follows on previous findings and elucidates for the first

time the role of Vangl and downstream signaling cascade in cytoneme formation and function in vertebrates systems. This study demonstrates that Wnt8a and Vangl overexpression reduces the number but increases the length and stability of Wnt expressing cytonemes. Wnt8a is also shown to activate the PCP pathway via Ror2 receptor and these act upstream JNK pathway activation. In addition, AGS gastric cancer cells transiently transfected with a Wnt/ β -catenin reporter showed an increasing Wnt pathway activation when co-culture with cells expressing Wnt8a/Ror2/Vangl2 suggesting that the Wnt8a/Ror2/Vangl2 derived cytonemes are functional. The study is cleverly designed and quantifications were carefully done. The findings are interesting and novel. The experimental data and mathematical model also provides compelling evidence that supports the majority of authors' claims.

There are however several concerns:

1) Could authors elucidate Δ N-Vangl2 function?

In the present work it is clear that Δ N-Vangl2 is a non-functional Vangl2. Although, initial studies suggest that Δ N-Vangl2 may work as a dominant-negative which does not seem to be the case in this study (see for example Fig3D,E- expression of Δ N-Vangl2 has the same effects as the control, although in Figure 2J,G- Vangl and Δ N-Vangl have very similar effects in cytonemes elongation and retraction rates).

2) As the present work is based on Vangl2 function, could authors validate some of the experimental observations by analyzing the zebrafish Vangl mutant or perform KD experiments using Vangl2 anti-sense oligo morpholinos?

3) Co-culture of AGS gastric cancer cells expressing 7xTCF-nls-mCherry with cells expressing Wnt8a/Vangl/ ROR2 (Fig5):

I have concerns that different combinations of Wnt/Vangl/Ror2 constructs may affect cell proliferation and therefore affect the read-out of Wnt activation in neighbour cells. Could authors provide evidence that at the time of the analysis the proportion of cells expressing the Wnt/Vangl/Ror2 related constructs versus 7xTCF-nls-mCherry expressing cells is similar in all conditions?

4) The mathematical model assumes that cytonemes were the exclusive transport mechanism for Wnt. Is it possible to assume another scenario in which, for example, cytonemes are shorter or absent but Wnt is still present? This would complete the scenario of long, normal (wt) and shorter cytonemes

Minor details:

i) When analysing cytoneme dynamics authors measured the "protrusions" and "retraction" rates. I suggest authors replace "protrusion" by "elongation" as this seems to better represent the process.

ii) Expression of Vangl2 led to significantly fewer but longer cytonemes per cell compared to control ¹¹_{SEP}- the branching was also abnormal. Could authors better explained / illustrate the abnormal branching?

iii) Figures:

- Change colour combination from red/green to magenta/green

- Improve figures' annotations – for example in Fig3b- it is difficult to see the object indicated by the yellow arrow. Cytonemes are very thin structures and difficult to see- increase image magnification when necessary, use drawings to help the reader.

	Major comments	
	Reviewer 1	
	This work from the Scholpp lab continues their impressive and insightful investigations into the way that Wnts are distributed by cytonemes. There is much to like and admire in this installment, especially in the way that it progresses from identification of a role for Vangl2 in cell culture, to a mathematical analysis that predicts how perturbations to cytonemes in response to changes in the Vangl2 pathway might affect development of zebrafish embryos, and most importantly to direct experimental tests of these predictions. I enthusiastically recommend it for publication. My suggestions for clarification/improvement are minor.	We thank the reviewer for the encouraging words. In the following we try to address all suggestions raised by the reviewer.
1	Figure 1 Legend, text, Methods do not describe many essential details such as number of samples, whether they are fixed or unfixed, how fluorescence intensity was scaled, whether cytoneme position is shown tip to cell or cell to tip, length variation, ...	We have added additional information to all figure legends. In detail, we included the sample for all experiments, how they were imaged (live/fixed) and how we measured fluorescent intensity from tip to cell.
2	Conditions for controls for transfections and cytoneme length measurements should be described better. Do cytonemes change at different times of culture, after exposure to transfection reagents?	We have checked the formation of cytonemes and found no differences in length or dynamic after exposure to transfection medium or at different time points of culture. We added this information to the Material and Methods part.
3	If I understand correctly, the cytonemes that are tabulated in Fig 2 are filopodia that contain Wnt8a. For the experiments in Fig 2, is the effect only these filopodia or all filopodia? Is cell viability, mitotic rate, or any other measure of cell vitality affected?"	In Fig. 2 we addressed the consequences on Wnt8a-positive filopodia - cytonemes. For the sake of completeness, we added the numbers for all filopodia in Supplementary data Fig. 3. We clarified the expression in the text.
4	Figure 3 Is the method for embryo analysis described? -Are these measures of intact embryos or dissociated cells from embryos? -How many fluorescent-positive cells did the embryos have? -How were the cells chosen for analysis? Are the controls uninjected or sham-injected? -Do I understand correctly that here the definition of cytonemes is not the subset of filopodia that contain Wnt? -To what extent are the data for cell culture and embryos consistent or inconsistent?	The required information has been added to Materials and Methods, Results, and to the Figure Legends. -High-resolution imaging was performed in the intact, living zebrafish embryo. -We generated clones of approx. 150 cells in an embryo at 5hpf. -The ctrl sample are micro-injected/ transfected with a ctrl construct. -We define cytonemes as Wnt8a bearing signalling filopodia. -Most of the data observed in TC experiments were similar to the ones observed in the embryo. However, one difference is very obvious: the number of cytonemes is slightly reduced in PAC2 cells after transfection with Vangl2 - not significantly. However, our in vivo experiments demonstrate that the number is unchanged in the zebrafish embryo.
5	Figure 4B,C Is the greater variation in cytoplasmic/nuclear distribution responsible for the apparent increase for the Ror2/Vangl2/Wnt experiments? What would violin plots look like?	The box & whiskers plot has been replaced by a violin (Fig. 4B). Based on this analysis we show that there is a similar variability in the different treatments.
6	Line 248 Please explain why the mouse intestinal crypt was chosen and is presumably a better system for the	We explain the importance of Wnt signalling in the intestinal crypt in the Results part and cite a few milestone papers.

	experiments on Wnt transport then the zebrafish embryo. Figure 6 Does this system provide the authors with the possibility of observing if the cytonemes are polarized in a particular direction or toward a particular field?	We agree with the reviewer that directionally is an important topic. Unfortunately, this system cannot be used to detect polarization of cytonemes. However, the main group of long Wnt8a cytonemes are directed towards the animal pole during zebrafish gastrulation (Stanganello et al., 2015; PMID: 25556612)
7	Figure 7 Can the authors provide a distance scale so we can understand the dimensions of Wnt signalling being modelled? Also, a developmental time scale so we have a context for the cytoneme protrusion/retraction rates?	We added more information (a distance scale, developmental timing) and expanded the description of the figure.
8	Figure 8 Can the authors provide some validation for the general state of these embryos – cell death/cell division/cell shape? Do they develop further?	Further information regarding embryo state and development has been provided in the Materials and Methods for the ISH analysis section
Reviewer 2		
	The manuscript by Brunt et al describes a finding that Vangl2 could regulate the formation of Wnt8-positive cytonemes (signaling filopodia) and thus have a positive function in the Wnt ligand dissemination and paracrine Wnt/β-catenin signaling. Although this story is interesting, the role of Vangl2 in regulating filopodia has been documented (Love et al, Development 2018) in a clean genetic model (Vangl2 trilobite mutant zebrafish) rather than by overexpression experiments mainly used in this study. I am actually very surprised that trilobite mutant was not analyzed in this study. The role of Vangl2 in initiating filopodia has also been demonstrated in mammalian cells (Shafer et al, Dev Cell 2012). In addition, the data presented are largely association, often time no definitive link can be established. There are some fundamental issues to be addressed, the specific points are listed below.	We reply to all the comments point-by-point in the following.
1	Obviously, the authors misunderstood the relationship between Ror2 and Vangl2 or they didn't read the literature carefully. They claimed that the kinase domain of Ror2 phosphorylates Vangl2 (page 3, line 68), however, there was no such evidence. Instead, many papers have shown that Vangl2/Vang is phosphorylated by casein kinase 1 (Gao et al, Dev Cell 2011; Kelly et al, Cell Rep 2016; Yang et al, Cell Res 2017; Strutt et al, Elife 2019).	We are grateful for the comment of the reviewer. In the text we now highlight the role for CK1δ/ϵ in phosphorylation of Vangl2. Furthermore, we cited the requested publication demonstrating the role for these kinases.
2	Page 3, line 72-73, they described that "During zebrafish gastrulation, Vangl2 is asymmetrically localised at the plasma membrane and localizes to growing filopodia²³". I could not find the picture in the cited reference regarding the filopodia localization of Vangl2, please point it out. In their own studies, the overexpressed Vangl2 could be observed in the cytoneme (Figure 1D). But according to the signal in the cell body, Vangl2 is everywhere rather than the normal localization on the cell surface. Also for the clustering of Vangl2 and Wnt8a in the cytoneme, how to exclude the artifact of overexpression? Is there any evidence showing the cytoneme localization of endogenous Vangl2, which is more physiologically relevant for Vangl2's function?	We agree with the reviewer that the overexpression experiments in Fig. 1 should not be used to define the localisation of Vangl2 without additional supporting evidence. To address this question, we tested several commercially available antibody in IHC staining with zebrafish Vangl2 in Pac2 cells, but we were not successful. Therefore, we used a human Vangl2 antibody in AGS cells - the same cell line we use in Fig. 4 to demonstrate the impact of Vangl2 cytonemes on signalling. We could demonstrate that human Vangl2 is localised to cytoneme tips of AGS cells (Suppl. Fig. 1.C). For direct comparison, we show Pac2 cells expressing Vangl2-GFP, showing a very similar localisation (Suppl Fig. 1D). Next, we used a high-resolution approach to visualise the localisation of Vangl2-GFP expressed under the Vangl2 promoter in zebrafish. Similar to our observation in Pac2 cells

		and AGS cells, we show that Vangl2 - expressed with near endogenous concentration - is localised to cytoneme tips (Fig. 3A,B). For further clarification, we add 3D rendering and 3D rotational movies (Supp. Movies 1&2). Furthermore, Reference no.25 (Love et al., 2018; PMID: 30327324) shows membrane accumulation of Vangl2 at sites where filopodia start to protrude and Reference no. 8 (Davey et al., 2016; PMID: 26990447) demonstrates Vangl2 accumulation at tips of retracting filopodia.
3	Figure 1H-L, how the fluorescent intensity among different samples are normalized? The number of filopodia (N=5) measured is too low to draw a firm conclusion. Figure 1D and 1J, when GFP-Vangl2 and Ror2-mCherry are co-expressed, a beads-on-string pattern could be observed. If that is the case, why fluctuation of signals along the cytoneme was not detected.	We agree with the reviewer that n=5 would be too low to come to a firm conclusion. Therefore, we measured the fluorescence along more cytonemes as specified in the figure legends: n= 17,12,12,13,11. Some fluctuations along the cytoneme could be detected see Fig. 1J. However, as these Vangl2/Ror2 clusters are distributed arbitrary along the cytonemes the averaging of n=12 displayed in Fig. 1J of several cytonemes (n=12) cancels out individual clusters. To strengthen our argument that Wnt is required for Ror2/Vangl2 interaction, we performed a fluorescence cross-correlation spectroscopy (FCCS) experiment. Exposure to Wnt5a or Wnt8a leads to cross-correlation - meaning close interaction of Vangl2 and Ror2.
4a	Page 5, line 128-129, they found “Activation of Vangl2 caused a 58.5% reduction of cytonemes per cell”. “The number of cytonemes are unaltered” by Ror2 transfection. For filopodia, Ror2 expression significantly increased the number of filopodia but Vangl2 decreased it (Supplementary Figure 3G). Why do Ror2 and Vangl2 have opposite effects on filopodia?	To address this question, we analysed the data set again and included more measurements for cytoneme length in vitro and in vivo. We find, indeed, that the number of cytonemes decreases in vitro – though not significantly, however, we do not observe a similar decrease in vivo. This observation is supported by the analysis of the Vangl2^{10A} mutant in vivo which demonstrates a significant reduction of the number of cytonemes in both settings. Similarly, the number of filopodia seems to be unaltered after Vangl2 expression, however, is significantly reduced after expression of the Vangl2^{10A} mutant. Therefore, we believe that Vangl2 is necessary but not sufficient to induce cytonemes. However, Vangl2 is triggering the formation of longer cytonemes.
4b	The Ror2/Vangl2 PCP signaling is specific for Wnt8-positive cytoneme? Are Ror2 and Vangl2 expressed in Wnt8-negative cytoneme?	We find that Wnt8a forms a complex of Ror2/Vangl2, which induces the formation of long cytonemes. However, there are 19 Wnt ligands in humans and 23 in fish so we cannot exclude that other Wnts may use a similar transport mechanism. AGS cells express prominently Wnt1 and Wnt3 and it seems that Vangl2 has a similar effect (Fig. 5). Furthermore, Wnt2b seems to be the crucial Wnt ligand expressed by telocytes in the intestinal crypt in mouse. Also, we find that Vangl1/2 is required for cytoneme formation. Therefore, we would argue that this is a more general mechanism.

5	Figure 2D and F, with two channels, how do they know the cell is positive for three overexpressed genes (Wnt8a, Vangl2 and Ror2)?	Based on our experience, we find that Pac2 cells either take up all constructs or none of them. We tested this with various combination of fluorescently marked constructs and only a very small proportion of cells (below 5%) show the expression of only one marker. Together with the striking changes of the phenotype we believe that we analysed cells positive for all constructs.
5b	Supplementary Figure 3A-F, the intensity is too low to see the length and number of cytonemes in these cells.	We are sorry for the loss of quality during the upload of the images. We increased the intensity and resolution of the images in Fig. 3C-G and hope that it is now possible to see the fine protrusions.
6	They further used N-terminal deletion mutant of Vangl2 (Δ N-Vangl2) lacking the two Ser/Thr phosphorylation clusters to block Vangl2-mediated signaling. When I looked into the reference they cited (Gao et al., 2011) and a further in vivo studies of Vangl2 phosphorylation (Yang et al. Cell Res 2017), I could not find any evidence to support their conclusion that Δ N-Vangl2 could block Vangl2-mediated signaling. These papers, particularly the later one, showed dominant negative effects of phospho-mutant Vangl2 which appeared to affect more than Vangl proteins. Why not knock out Vangl1 and Vangl2 in the PAC2 cells to have more clean and convincing data, or at least use phospho-mutant Vangl2.	We are grateful for this comment and we repeated the experiments with the all-phospho Vangl2 mutant S5~17A::S76~84A (Gao et al., 2011 PMID: 21316585; Yang et al., 2017; PMID: 29056748), in which 10 Ser/Thr are replaced by Ala (Vangl2 ^{10A}). Gao et al., 2011; Yang et al., 2017 have used this mutant as a tool to block Vangl2-mediated signalling. We used this mutant form of Vangl2 in PAC2 cells and in the zebrafish embryos. Indeed, the experiments using this mutant confirmed our hypothesis that Vangl2 is necessary but not sufficient for cytoneme formation. Furthermore, Vangl2 ^{10A} shows an even stronger phenotype compared to Δ N-Vangl2. We show these new results in Fig. 2, Fig. 3, Fig. 8.
7	Figure 3, studies in zebrafish embryos. As mentioned earlier, the Vangl2 trilobite mutant zebrafish has been widely used for PCP studies. This is the most reliable way to study Vangl2's function in cytoneme.	We thank the reviewer for this comment - however, we preferred to use the Vangl2 ^{10A} mutant construct and found this to be a reliable tool to block Vangl2 signalling also in the embryo - see Fig. 3 and Fig. 8.
8	Figure 3A-C, the signal intensity is much higher in B compared to A and C. They need to be shown at similar level. I can argue that the longer cytonemes in B are simply due to the higher overall fluorescent intensity.	All images were taken with same settings and intensity level have been kept at the same level.
9	Figure 4A, B. To be consistent with previous experiments, PAC2 cells should be used for KTR reporter experiment?	In this experiment, we used HEK283T cells. Due to technical reasons, HEK283T have a much higher transfection efficiency (ca 90%) compared to PAC2 cells (ca 30%). In this experiment, it is important that producing cells and receiving cells are in close proximity. Therefore, we have chosen a cell line which would ensure a good exchange of signal via cytonemes.
9b	What's the effect of Wnt5a and Wnt8a alone?	As requested, we added the data for Wnt5a and Wnt8a. Finally, we display the data set as violin plots as requested by reviewer 3.
10	Figure 4F, it seems the length of cytonemes decreased over the time in the Vangl2 group and it is similar to the pattern of Vangl2 + SP group. How to explain this?	We performed more repeats for this experiment of measuring filopodia length of Vangl2 positive cells over 120 minutes. Even though there is some reduction in filopodia lengths over time, this is not significant in the non-drug group.
11	Figure 5A-C, the Wnt signaling activity experiment. This is only an association between the upregulation of Wnt signaling and Vangl2-regulated cytonemes. I can argue that this is due to the increased secretion of Wnt proteins somehow triggered by the Wnt8a/Ror2/Vangl2 overexpression in source cells or even indirectly through other mechanisms, for example, Vangl2 or Ror2 might	We are grateful to the reviewer for this insightful question. To strengthen our argument that Vangl2 function increases cytoneme length and thus enhances Wnt transport and signalling, we complemented the experiments with a tool in which we block filopodia formation by other means - and therefore, independent of PCP signalling. Here, we

	stabilize Wnt8a or promote maturation of the ligand, why necessary through cytonemes?	use the IRSp53 ^{4K} mutant, which has been shown in various systems viz. tissue culture and zebrafish to reduce significantly the formation of filopodia including cytonemes. In Fig. 2 and Fig. 3 we demonstrate the IRSp53 ^{4K} reduces significantly the number of Wnt8a/Vangl2-induced cytonemes in vitro (Fig. 2) and in vivo (Fig. 3). Wnt/Vangl2/IRSp53 ^{4K} expression leads to a significant reduction of paracrine Wnt signal induction (Fig. 5, Fig. 6). We further simulated the effect of this parameter in Fig. 7 and could confirm the prediction of the simulation in Fig. 8.
11b	More importantly, if their model is true, does Vangl2 mutant animals (drosophial, zebrafish, or mouse) show the downregulation of canonical Wnt signaling? The in vivo evidence will be very critical for the conclusion.	To provide further evidence that Vangl2 function is required for the dissemination of Wnt8a also in the embryo, we impaired Vangl2 function via injection of the all-phospho Vangl2 mutant Vangl2 ^{10A} . We find that blockage of Vangl2 function during zebrafish gastrulation leads to a significant reduction in cytoneme number (Fig. 3). We further show that expression of the Vangl2 ^{10A} mutant lead to a partial rescue of Wnt8a overexpression in vivo , suggesting that cytonemes are important for paracrine Wnt8a signalling (Fig. 8).
12	Figure 6B, C, “the number of filopodia was significantly reduced after Vangl1 knockdown and even more markedly reduced after double knockdown of Vangl1/Vangl2”. But as shown earlier in supplementary Figure 3G, the number of filopodia by Vangl2 overexpression is also reduced. How to explain this discrepancy?	We repeated the experiments and indeed, we find a downregulation of filopodia in the in vitro experiments in (Fig.2) – though not significantly. However, in the in vivo experiments we found that Vangl2 expression does not alter the number of cytonemes (Fig. 3). Furthermore, Vangl2 blockage by the all-phospho mutant Vangl2 ^{10A} leads to a strong reduction of numbers. We interpret these results that Vangl2 is necessary to for cytoneme formation, but not sufficient.
13	Figure 6E, F. Fewer organoids formation is not a direct readout of downregulated Wnt signaling. Few organoids could be due to many reasons, for example, PCP may be important for telocyte’s normal function or morphology. Again, this is only an association.	We are grateful for the possibility to further explain the background and our experimental strategy. It is well established that the function of the stroma in the mouse organoid assays is to provide Wnts, RSPO, an EGF ligand, and a BMP inhibitor (Sato, 2009, PMID: 19329995; Farin, 2012, PMID: 22922422; Kabiri 2014, PMID: 24821987). The Clevers lab (PMID: 222922422) showed that deletion of WNT3 from organoids prevents their growth, and that can be rescued by addition of over-expressed Wnts from feeder cells that are not telocytes. The Virshup lab showed (Kabiri et al. 2014; PMID: 24821987) that organoids that can’t make Wnts are rescued by telocytes or by L cells or HEK293 cells making WNT3A. In our culture medium we provide EGF ligand, RSPO1, and a BMP inhibitor. Hence, the only remaining function of the stroma is provision of Wnt ligand. We, therefore, respectfully disagree that this is only an association. The data are most consistent with the VANGL1/2 knockdown impairing Wnt delivery to the intestinal stem cells. In particular, in these experiments, we co-cultivate telocytes with crypt cells, which are not able to produce a Wnt signal on their own as they carry a

		mutation for PORCN. These crypt cells depend on the supply of external Wnts e.g. co-cultivated Wnt3a positive L cells or WT myofibroblasts, which we have shown recently (Greicius G et al., 2018; PMID: 29559533, Mattes et al., 2018; PMID: 30060804). In this sophisticated experiment, we can now directly evaluate the dependency of Wnt protein supplied by the stroma i.e. telocytes. If we block Vangl1/2 in telocytes, they form significantly less cytonemes and Vangl1/2 KD telocytes are less able to rescue the survival of crypt organoids. To demonstrate that telocytes have a normal phenotype and engage with the crypt cells, we added high-resolution images of the 3D co-cultivated organoids. To further strengthen our argument that cytonemes are essential for Wnt trafficking from telocytes to the crypt cells, we knocked-down IRSp53 function in telocytes - additional data in Fig. 6. Indeed, we find a reduction of cytonemes, and a reduced ability to support organoids. In summary, the data support the model that Vangl1/2-regulated cytonemes are essential for Wnt delivery from telocytes to the crypt cells and thus for organoid survival.
14	Figure 7, the authors tested cytoneme model, how about the diffusion model? If Wnt ligand transport is independent of cytonemes and just relies on secretion and diffusion, is there any difference compared to cytoneme simulation? If does, how to distinguish them experimentally?	In the meantime, we have compared a cytoneme-based transport mechanism to a diffusion-based one in a separate publication (Rosenbauer et al., 2020). We added a paragraph regarding these findings to the discussion.
15	Figure 8, “Vangl2 activity in the Wnt source cells regulates neural plate patterning in zebrafish neurogenesis” is an overstatement. It is possible that the change of expression patterns of these markers is secondary to a CE defect.	We thank the reviewer for the comments and changed this statement in the text and talk now about an 'influence' instead of 'regulation'. In summary, we analysed the effect of Vangl2/Wnt8a on posterization via β -catenin signalling and convergence & extension PCP signalling in Fig. 8C-E. We find that overexpression of Wnt8a/Vangl2 show a posterization (reduction of forebrain tissue) - an indication of β -catenin signalling (Fig. 8D). Furthermore, we are not aware of any experimental situation in which PCP signalling alteration leads to a posteriorization. However, in addition, we find that PCP signalling is altered as we find a broader neural tube - hence a CE defect (Fig. 8E).
15b	Since overexpression Vangl2 is well known to cause CE defect, how did they distinguish the direct effect of Vangl2 on PCP and indirect effect through cytoneme on canonical Wnt signaling?	We repeated the experiments and overexpressed Wnt8a/Vangl2 and, in parallel, blocked cytoneme formation independently by IRSp53 ^{4K} mutant. Indeed, we find a rescue of the phenotype, suggesting that Vangl2 influences the dissemination of Wnt8a protein by cytonemes. (Fig. 8)
15c	Please clearly specify the region of Wnt source cells and receiving cells, is the phenotype restricted to the receiving cells? This part of data is not convincing, difficult to correlate it with cytoneme.	We further blocked Vangl2 function by using the Vangl2 ^{10A} mutant and indeed we do not observe posteriorization of the expression domains of gbx1 or the ntl (Fig. 8)

	How Wnt proteins can spread in the extracellular space is an important but also controversial topic. A very recent study excluded all the previously proposed mechanisms of Wnt transport (e.g., lipoprotein particles, exosomes, chaperone) and demonstrated a new mechanism via glypican shielding (McGough et al, Nature 2020). However, whether cytoneme is indeed essential for Wnt dissemination is unclear due to the lack of genetic tool and evidence. Given the tremendous efforts made by this group on the Wnt cytoneme in the past, convincing genetic evidence is expected.	We are grateful for the comment of the reviewer who clearly appreciate the importance of this topic. Indeed, in the McGough paper several transport mechanisms i.e. lipoprotein particles, exosomes & transport proteins have been excluded - except cytonemes (McGough et al., 2020; PMID: 32699409). We agree, it will be interesting to analyse the link between Glypicans, cytonemes and Wnt trafficking in the future.
	Reviewer 3	
	The m/s by Brunt et al focusses on the role of cytonemes and their regulation by Vangl/Wnt signalling. The authors show a role for Vangl2 in Wnt distribution in zebrafish fibroblasts in culture and in vivo in gastrula stage embryos. In addition, Vangl2 activation appears to result in longer and fewer projections in zebrafish and in cultured human gastric cancer cells and mouse intestinal telocytes. Reduced Vangl leads to shorter cytonemes and reduced canonical Wnt signalling. The authors then modelled the function of Vangl2 in zebrafish gastrulation and predicted a shift of the morphogenetic gradient and altered neural tissue patterning, and tested these on the zebrafish neural plate. Overall, the manuscript reports interesting findings. However, the current data do not fully support the conclusions. Some additional analysis is needed to conclusively demonstrate a role for cytonemes in Vangl/Wnt signalling in cultured cells and in embryos.	We thank the reviewer for the motivating review, and we hope we could address all suggestions raised in their report.
1	With regard to cytoneme contacts between cells: In some instances, it is hard to discern if the contacts are actually there between cells. Showing 3D rotations of cytoneme contacts between two cells would be more conclusive (e.g. for Figure 3 I).	We added 3D rotations to the following data sets to illustrate changes to cytoneme contact number and branching: Wnt8a/mem-mCherry, Wnt8a/Vangl2, Wnt8a/Vangl2^{10A}, Wnt8a/Vangl2/IRSp52^{4K} (Supp. Movie 3-6).
2	There are only 1-2 cytonemes per cell. That being the case, it is difficult to assess a change from 2 to 1 (e.g. in DN-Vangl expressing cells,; Figures 2 and 3). Because the cytoneme N's are not meaningful and length ranges from 5-20 um, an alternate way of assessing contacts is required	We agree with the reviewer that the difference in number seems to be small. However, we would like to bring into consideration that this is in fact a change of 100%. Furthermore, one needs to take into account that the length of the cytonemes nearly doubles when Vangl2 is overexpressed. We further agree that the way we measured the contacts - via protrusion and retraction time - is not ideal. Therefore, we quantified the number of contacts per cytoneme in vivo (Fig. 3). We find that cytonemes form also more contacts - a beautiful example for multiple contacts is shown in Fig. 3D. So, we believe that the changes of cytoneme behaviour (number, length & contacts) is meaningful, demonstrated by the changes in signalling in Fig. 5 & 6 and tissue patterning in Fig. 8
3	The perdurance of individual contacts could be a better measure of stability. For instance, to demonstrate that cytoneme stability is regulated by Vngl2, in addition to	We agree with the reviewer fully. Quantification of the duration of the contact would be the optimal way. However, due to technical reasons it is very

	length and numbers per cell, the authors should assess the DURATION of each contact. If the cytonemes are severed at a different point along the projection length (from the contact location) that can be indicated, and regardless of breakpoint location, the time of severing (before cytoneme retraction) can be taken as the end point of contact.	difficult to quantify this parameter in the living zebrafish embryo. This is due to the high dynamic and the fragility of cytonemes as well as the cell movement and tissue movement in the zebrafish embryo. We, therefore, quantified the number of contacts per cytoneme - see comment above - as a reliable parameter, which can be correlated with signalling intensity.
4	Fig 4b v and vi show cells with both nuclear and cytoplasmic expression or cytoplasmic expression alone (of the JNK signalling reporter).	We agree with the reviewer that KTR-mCherry localisation can still be seen in the nucleus after PCP activation. This has also been observed in the original publication (Regot, et al., 2014; PMID: 24949979 Miura et al., 2018; PMID: 30184500). However, the ratio shifts from a high nucleus/weak cytoplasm localisation to a weak nucleus/high cytoplasmic localisation. We have quantified the ratio in detail in Fig. 4B. Based on the suggestions from Reviewer 1 we show the distribution in a violin plot to visualize the spreading of the data sets.
4b	These should be discussed and the numbers expressing nuclear, cytoplasmic and both should be shown. In addition, some cells (Fig 4 iv and vii) appear multinucleate. Why is that so? A membrane marker would be useful here.	We have added the missing experiments i.e. treatment with Wnt proteins. The cells are not multinucleated, rather they form clusters.
5	Figure 6: The telocytes where Vangl1 or 1+2 are knocked down show a cell shape change from spindle to more lobed morphology. Perhaps this underlies the defect in organoid formation in co-cultures with intestinal epithelial cells, rather than arising from cytoneme contact defects. How do the authors distinguish cytoneme defects from general morphological defects in these experiments? Other manipulations that lead to cell shape change (without affecting contacts) could be used as controls.	We appreciate the comment of the reviewer. The primary telocytes isolated from mouse intestine in our cultures are very heterogeneous in shape, which may simply explain why the figure has some variation in structure of individual cells. When Vangl1/2 are knocked down there is a clear change in cytonemes, as described, but we have not detected a consistent change in other aspects of cell shape. We also think the inability of the Vangl-knockdown telocytes to support organoids is most likely due to a cytoneme contact defect rather than a shape-change defect, because we ensure robust contacts between the telocytes and the organoids in the way we set up the experiment. The telocytes and the crypts are mixed together and then gently centrifuged (400 x g for 5 min) together prior to resuspension in a small volume of Matrigel for plating. We arrived at this protocol because a simple plating of crypts on top of telocytes is not very efficient. As can be seen in Fig. 6D, both wildtype and Vangl1/2 knockdown telocytes form extensive contacts with organoids with this protocol. Based on this data we do not think cell shape change affecting telocyte-organoid contact is playing a role.
6	-The modelling of neural plate patterning (Figure 7) and test of the model in embryos (Figure 8) is interesting. The data presented show a role for vangl and wnt8 in hindbrain gbx1 expression and neural plate patterning; however, these do not conclusively show that the effect of vangl/wnt8 is cytoneme dependent. A more direct/local perturbation of cytonemes and its effects on Wnt distribution is needed to demonstrate this link.	We are grateful to the reviewer for this insightful question. To strengthen our argument that Vangl2 function increases cytoneme length and thus enhances Wnt transport and signalling, we complemented the experiments with a tool in which we block filopodia formation by other means - and therefore, independent of PCP signalling. Here, we use the IRSp53^{4K} mutant, which has been shown in various systems viz. tissue culture and zebrafish to reduce significantly the formation of filopodia

		including cytonemes. In Fig. 2 and Fig. 3 we demonstrate the IRSp53 ^{4K} reduces significantly the number of Wnt8a/Vangl2-induced cytonemes in vitro (Fig. 2) and in vivo (Fig. 3). Wnt/Vangl2/IRSp53 ^{4K} expression leads to a significant reduction of paracrine Wnt signal induction (Fig. 5, Fig. 6). We further simulated the effect of this parameter in Fig. 7 and could confirm the prediction of the simulation in Fig. 8.
6b	This is important because the movies of embryos don't show as marked a change in vivo as the cultured cells e.g., upon DN-Vangl injections (e.g., Movie 5 versus 8).	We agree with the reviewer that the DN-Vangl2 is a weak tool to block Vangl2 signalling. We, therefore, repeated the experiments with the all-phospho Vangl2 mutant S5~17A::S76~84A (Gao et al., 2011; Yang et al., 2017), in which 10 Ser/Thr are replaced by Ala (Vangl2 ^{10A}). Gao et al., 2011; Yang et al., 2017 have used this mutant as a tool to block Vangl2-mediated signalling. We used this mutant form of Vangl2 in PAC2 cells and in the zebrafish embryos. Indeed, the experiments using this mutant confirmed our hypothesis that Vangl2 is necessary but not sufficient for cytoneme formation. Furthermore, Vangl2 ^{10A} shows an even stronger phenotype compared to ΔN-Vangl2. We show these new results in Fig. 2, Fig. 3, Fig. 8.
7	In many cells, Wnt8 is seen not only at the tips but also along the length of the projections (e.g. Fig 1). These should be indicated with smaller arrowheads, to distinguish from the ones at the cell tips.	We added smaller arrowheads to Wnt8a localisation along the cytonemes i.e. Fig. 1, 2, 3.
7b	Figure 4A; typo - change activity to activity	We corrected this mistake.
7c	Check and correct the manuscript for syntax/grammar and typographical errors.	We checked and corrected the manuscript for syntax/grammar and typos.
Reviewer 4		
	Long distance communication mediated by filopodia-like protrusions is an emerging field. There is increasing evidence that it plays important roles in tissue patterning by presenting ligands or receptors to distant cells. Cytoneme is one of the cellular processes shown to carry Wnt8a and to activate the pathway and pattern the neural tissue. Cytoneme formation has been associated with Wnt8a-mediated Ror2 activation. The current study follows on previous findings and elucidates for the first time the role of Vangl and downstream signaling cascade in cytoneme formation and function in vertebrates systems. This study demonstrates that Wnt8a and Vangl overexpression reduces the number but increases the length and stability of Wnt expressing cytonemes. Wnt8a is also shown to activate the PCP pathway via Ror2 receptor and these act upstream JNK pathway activation. In addition, AGS gastric cancer cells transiently transfected with a Wnt/β-catenin reporter showed an increasing Wnt pathway activation when co-culture with cells expressing Wnt8a/Ror2/Vangl2 suggesting that the Wnt8a/Ror2/Vangl2 derived cytonemes are functional. The study is cleverly designed and quantifications were carefully done. The findings are interesting and novel. The experimental data and mathematical model also	We are grateful for the uplifting comments to our manuscript and in the following, we addressed all comments point-by-point-

	provides compelling evidence that supports the majority of authors' claims.	
1	Could authors elucidate Δ N-Vangl2 function? In the present work it is clear that Δ N-Vangl2 is a non-functional Vangl2. Although, initial studies suggest that Δ N-Vangl2 may work as a dominant-negative which does not seem to be the case in this study (see for example Fig3D,E- expression of Δ N-Vangl2 has the same effects as the control, although in Figure 2J,G- Vangl and Δ N-Vangl have very similar effects in cytonemes elongation and retraction rates).	We are grateful for this comment and we repeated the experiments with the all-phospho Vangl2 mutant S5~17A::S76~84A (Gao et al., 2011 PMID: 21316585; Yang et al., 2017; PMID: 29056748), in which 10 Ser/Thr are replaced by Ala (Vangl2 ^{10A}). Gao et al., 2011; Yang et al., 2017 have used this mutant as a tool to block Vangl2-mediated signalling. We used this mutant form of Vangl2 in PAC2 cells and in the zebrafish embryos. Indeed, the experiments using this mutant confirmed our hypothesis that Vangl2 is necessary but not sufficient for cytoneme formation. Furthermore, Vangl2 ^{10A} shows an even stronger phenotype compared to Δ N-Vangl2. We show these new results in Fig. 2, Fig. 3, Fig. 8.
2	As the present work is based on Vangl2 function, could authors validate some of the experimental observations by analyzing the zebrafish Vangl mutant or perform KD experiments using Vangl2 anti-sense oligo morpholinos?	We believe that we have validated the experiments by confirming our data with the Vangl2 ^{10A} mutant - see comments above.
3	Co-culture of AGS gastric cancer cells expressing 7xTCF-nls-mCherry with cells expressing Wnt8a/Vangl/ ROR2 (Fig5): I have concerns that different combinations of Wnt/Vangl/Ror2 constructs may affect cell proliferation and therefore affect the read-out of Wnt activation in neighbour cells. Could authors provide evidence that at the time of the analysis the proportion of cells expressing the Wnt/Vangl/Ror2 related constructs versus 7xTCF-nls-mCherry expressing cells is similar in all conditions?	We agree with the reviewer that cell proliferation could have an impact in that experiment. To exclude this issue, DAPI staining was used to ensure equal numbers of cells were present in each frame analysed to rule out an influence of cell proliferation. We are sorry that this was not clearly mentioned in the text and we have added this information to the Material & Methods part.
4	The mathematical model assumes that cytonemes were the exclusive transport mechanism for Wnt. Is it possible to assume another scenario in which, for example, cytonemes are shorter or absent but Wnt is still present? This would complete the scenario of long, normal (wt) and shorter cytonemes	We thank the reviewer for this comment. Indeed, we have added a scenario in which the number of Vangl2 positive cytonemes significantly reduced. This simulation mirrors the observation of expression of IRSp53 ^{4K} mutant in the Wnt8a/Vangl2 background - see Fig. 2 & 3. We are pleased to report that this treatment (partially) rescues the effect of longer Vangl2 cytonemes in terms of signalling (Fig. 5) and tissue patterning (Fig. 8).
5	i) When analysing cytoneme dynamics authors measured the "protrusions" and "retraction" rates. I suggest authors replace "protrusion" by "elongation" as this seems to better represent the process.	Due to technical limitations, we have replaced this analysis by quantification of the number of contact points per cytoneme. This is a better, easier, and more robust way - in addition it can be directly correlated with signalling strength.
5b	ii) Expression of Vangl2 led to significantly fewer but longer cytonemes per cell compared to control - the branching was also abnormal. Could authors better explained / illustrate the abnormal branching?	We visualized branching and increase of contact sites in Fig. 3. Furthermore, we used Imaris to generate rotation movies demonstrating the extend of arborization upon Vangl2 expression (Supplementary Movies 3-6)
5c	iii) Figures: Change colour combination from red/green to magenta/green	We have changed the colour regime and show b/w pictures when possible.
5d	Improve figures' annotations – for example in Fig3b- it is difficult to see the object indicated by the yellow arrow. Cytonemes are very thin structures and difficult to see- increase image magnification when necessary, use drawings to help the reader.	We have improved the presentation of the figures i.e. Fig. 1,2,3.

--	--	--

REVIEWERS' COMMENTS

Reviewer #1 (Remarks to the Author):

Reviewer suggestions have been addressed.

Reviewer #2 (Remarks to the Author):

This revised manuscript has been considerably improved, and most of my questions are now addressed. A few remaining concerns:

1. The endogenous Vangl2 localization in cytoneme is still not convincing (Suppl. Fig. 1C). Besides the few signals in the cytoneme, there are very strong signals in the cell body rather than the cell surface where Vangl2 is normally localized. This is strange. The exogenous GFP-Vangl2 also showed strong localization in the cell body without clear membrane localization (Suppl. Fig. 1D, and Fig 3A, B). As GFP may interfere with the Vangl2 transportation, authors may consider to express/inject a low dose of Vangl2 with a small tag.

2. The use of true dominant negative Vangl2 (Vangl2-10A) is nice, however, validation of some of key data using Vangl2 KO zebrafish is required to consolidate the conclusion, particularly, no previous study of Vangl2 mutants in any model organism showed downregulation of canonical Wnt signaling. Instead, some studies proposed Vang/Stbm could inhibit canonical Wnt signaling (e.g., Part and Moon, NCB 2001). Although this may be explained by cell autonomous vs non-autonomous effect, to have a firm conclusion, authors should analyze KO zebrafish or at least KD zebrafish using morpholino (as suggested by Reviewer #4). Vangl2 morpholino was reported to be very efficient and this can be tested quickly.

Reviewer #3 (Remarks to the Author):

The revised m/s addresses the key concerns and comments. The 3D movies (3-6) are useful and help show the contacts between cells better. The authors have tried to quantitate contacts between cells in addition to cytoneme numbers and length. This is not quite the same as duration of contacts which was suggested, but which the authors were unable to measure owing to technical reasons such as fragility of the cytonemes etc. The authors also used the IRSp534K mutant to manipulate Vangl2 function in cytoneme length independent of filopodia formation and PCP signalling, and an all phospho-Vangl2 mutant to block signalling to demonstrate that Vangl2 is necessary but not sufficient for cytonemes. As such the revised ms has improved substantially.

Reviewer #4 (Remarks to the Author):

The last version of the manuscript is much improved. Brunt et al. successfully addressed/reply to all my previous concerns. The use of IRSp53(4k) was also a good new addition to this study.

Reviewer #1 (Remarks to the Author): Reviewer suggestions have been addressed.	We are grateful for the positive response of Reviewer #1
Reviewer #2 (Remarks to the Author): This revised manuscript has been considerably improved, and most of my questions are now addressed. A few remaining concerns:	We are grateful for the positive response of Reviewer #2
1.The endogenous Vangl2 localization in cytoneme is still not convincing (Suppl. Fig. 1C). Besides the few signals in the cytoneme, there are very strong signals in the cell body rather than the cell surface where Vangl2 is normally localized. This is strange. The exogenous GFP-Vangl2 also showed strong localization in the cell body without clear membrane localization (Suppl. Fig. 1D, and Fig 3A, B). As GFP may interfere with the Vangl2 transportation, authors may consider to express/inject a low dose of Vangl2 with a small tag.	As requested in the first revision of the manuscript, we demonstrate sub-cellular localization of Vangl2 in various cell types and contexts: we show localization of GFP-Vangl2 in Pac2 cells in vitro (Fig. 1,2) and in zebrafish embryonic cells in vivo (Fig. 3). Furthermore, we added images showing the localization of GFP-Vangl2 expressed under the endogenous promoter in zebrafish embryonic cells in vivo (Fig. 3 – incl. 3D rendering) and we added an antibody staining of endogenous Vangl2 in AGS cells in vitro (Suppl. Fig. 1). Due to the low concentration of Vangl2-GFP in protrusions, we needed to increase the laser power and gain during imaging leading to a stronger signal in the cytosol. In support of our data, intracellular and membranous localization of Vangl2 was demonstrated in zebrafish (Sittaramane et al., 2013) and the rat cochlea (Montcouquiol et al., 2006). In conclusion, we and others find Vangl2 at the membrane and in the cytosol. Therefore, we suggest that Vangl2 is localized at cytonemes, at the plasma membrane and in intracellular vesicles, depending on the cell type and the tissue context.
2.The use of true dominant negative Vangl2 (Vangl2-10A) is nice, however, validation of some of key data using Vangl2 KO zebrafish is required to consolidate the conclusion, particularly, no previous study of Vangl2 mutants in any model organism showed downregulation of canonical Wnt signaling. Instead, some studies proposed Vang/Stbm could inhibit canonical Wnt signaling (e.g., Part and Moon, NCB 2001). Although this may be explained by cell autonomous vs non-autonomous effect, to have a firm conclusion, authors should analyze KO zebrafish or at least KD zebrafish using morpholino (as suggested by Reviewer #4). Vangl2 morpholino was reported to be very efficient and this can be tested quickly.	We thank the reviewer for this comment. To interfere with Vangl2 function, we reduced or increased Vangl2 function by various means (Fig. 2,3,4, 5, and 6), and we find that Vangl2 is required for cytoneme formation. In the first version of the manuscript, we used the N-terminal truncated form of Vangl2. Here, Vangl2 lacks N-terminus, including the phosphorylation sites, which are required for downstream activation. However - as Reviewer #2 pointed out – the Vangl2 (10A) mutant is a more specific tool to interfere with Vangl2 function (Gao et al., 2011, Yang et al., 2017). Therefore, we repeated the experiments with this mutant - with a similar outcome. Usage of this tool has been welcomed by all reviewers – specifically Reviewers #3. Besides, we knocked-down Vangl1/2 in telocytes by a siRNA approach (Fig. 6). If we block Vangl1/2 in telocytes, they form significantly fewer cytonemes, and Vangl1/2 KD telocytes are less able to rescue the survival of crypt organoids. All these experiments demonstrate that Vangl2 is required for Wnt cytoneme formation. Reviewer #2 suggests using the Vangl2 mutant and/or a Morpholino-based knock-down approach to confirm our findings further. The usage of the Vangl2 mutant is problematic. Our experimental strategy demands an analysis of the cytoneme-generating cells without interfering with the cytoneme-receiving cells. However, in the mutant, all cells are Vangl2 negative. Therefore, we would need to perform transplantation experiments, which are extremely difficult and if successful – only in 1 out of 4 cases the grafted cells would be Vangl2^{-/-}. This

	could raise concerns as the number of experiments would not be sufficient for statistical analysis. Similarly, a Morpholino-based knock-down approach is also very problematic. A Morpholino-based knock-down experiment can cause multiple off-target effects and thus requires numerous controls – even for a published Morpholino sequence – the added value of such an investigation would be limited. Nevertheless, a detailed discussion of alternative means to interfere with Vangl2 function has been added to the discussion section (p. 14)
Reviewer #3 (Remarks to the Author): The revised m/s addresses the key concerns and comments. The 3D movies (3-6) are useful and help show the contacts between cells better. The authors have tried to quantitate contacts between cells in addition to cytoneme numbers and length. This is not quite the same as duration of contacts which was suggested, but which the authors were unable to measure owing to technical reasons such as fragility of the cytonemes etc. The authors also used the IRSp534K mutant to manipulate Vangl2 function in cytoneme length independent of filopodia formation and PCP signalling, and an all phospho-Vangl2 mutant to block signalling to demonstrate that Vangl2 is necessary but not sufficient for cytonemes. As such the revised ms has improved substantially.	We are grateful for the positive response of Reviewer #3
Reviewer #4 (Remarks to the Author): The last version of the manuscript is much improved. Brunt et al. successfully addressed/reply to all my previous concerns. The use of IRSp53(4k) was also a good new addition to this study.	We are grateful for the positive response of Reviewer #4